# Endosomal egress and intercellular transmission of hepatic ApoE-containing lipoproteins and its exploitation by the hepatitis C virus

**Minh-Tu Pham[1,2], Ji-Young Lee[1,2], Christian Ritter[3], Roman Thielemann[3¤], Janis Meyer[3], Uta Haselmann[1], Charlotta Funaya[4], Vibor Laketa[2,5], Karl Rohr[3], Ralf Bartenschlager** [1,2,6] *

1 Department of Infectious Diseases, Molecular Virology, Center for Integrative Infectious Diseases Research, Heidelberg University, Heidelberg, Germany, 2 German Center for Infection Research (DZIF), Partner Site Heidelberg, Heidelberg, Germany, 3 BioQuant Center, IPMB, Biomedical Computer Vision Group, Heidelberg University, Heidelberg, Germany, 4 Electron Microscopy Core Facility (EMCF), Heidelberg University, Heidelberg, Germany, 5 Department of Infectious Diseases, Virology, Center for Integrative Infectious Diseases Research, Heidelberg University, Heidelberg, Germany, 6 Division Virus-Associated Carcinogenesis, German Cancer Research Center (DKFZ), Heidelberg, Germany

¤ Current address: Nordisk Foundation Center for Basic Metabolic Research, Faculty of Health and Medical Sciences, University of Copenhagen, Copenhagen, Denmark
* Ralf.Bartenschlager@med.uni-heidelberg.de

**Data Availability Statement:** All data are included in the manuscript and supporting information.

## Abstract

Liver-generated plasma Apolipoprotein E (ApoE)-containing lipoproteins (LPs) (ApoE-LPs) play central roles in lipid transport and metabolism. Perturbations of ApoE can result in several metabolic disorders and ApoE genotypes have been associated with multiple diseases. ApoE is synthesized at the endoplasmic reticulum and transported to the Golgi apparatus for LP assembly; however, the ApoE-LPs transport pathway from there to the plasma membrane is largely unknown. Here, we established an integrative imaging approach based on a fully functional fluorescently tagged ApoE. We found that newly synthesized ApoE-LPs accumulate in CD63-positive endosomes of hepatocytes. In addition, we observed the co-egress of ApoE-LPs and CD63-positive intraluminal vesicles (ILVs), which are precursors of extracellular vesicles (EVs), along the late endosomal trafficking route in a microtubule-dependent manner. A fraction of ApoE-LPs associated with CD63-positive EVs appears to be co-transmitted from cell to cell. Given the important role of ApoE in viral infections, we employed as well-studied model the hepatitis C virus (HCV) and found that the viral replicase component nonstructural protein 5A (NS5A) is enriched in ApoE-containing ILVs. Interaction between NS5A and ApoE is required for the efficient release of ILVs containing HCV RNA. These vesicles are transported along the endosomal ApoE egress pathway. Taken together, our data argue for endosomal egress and transmission of hepatic ApoE-LPs, a pathway that is hijacked by HCV. Given the more general role of EV-mediated cell-to-cell communication, these insights provide new starting points for research into the pathophysiology of ApoE-related metabolic and infection-related disorders.

**Funding:** This work was supported by grants from the Deutsche Forschungsgemeinschaft (DFG, German Research Foundation) – Project Number 272983813 – TRR 179 to RB and Project Number 240245660 - SFB1129 to RB and KR. The work was also supported by the German Center for Infection Research (DZIF), project number TTU 05.821 to RB. The funders had no role in study design, data collection and analysis, decision to publish, or preparation of the manuscript.

**Competing interests:** The authors have declared that no competing interests exist.

## Author summary

The post-Golgi egress pathway of hepatocyte-derived ApoE-containing lipoproteins (ApoE-LPs) is largely unknown. By using integrative imaging analyses, we show that ApoE-LPs are enriched in CD63-positive endosomes suggesting that these endosomes might be a central hub for the storage of ApoE-LPs from which they are released into the circulation. In addition, we provide evidence for the co-egress of ApoE-LPs with extracellular vesicles (EVs) along the late endosomal route and their transfer from cell to cell. This pathway is hijacked by the hepatitis C virus that induces the production of ApoE-associated EVs containing viral RNA. Given the important role of ApoE in multiple metabolic, degenerative, and infectious diseases, and the role of EVs in cell-to-cell communication, these results provide important information how perturbations of ApoE might contribute to various pathophysiologies.

## Introduction

Hepatocytes play a central role in lipid metabolism, both by production and clearance of plasma lipoproteins (LPs). Changes in hepatic lipid metabolism may contribute to chronic liver disease, such as nonalcoholic fatty liver disease [1]. Moreover, infections with hepatotropic viruses, most notably the hepatitis C virus (HCV), perturb hepatic lipid homeostasis, leading to hepatosteatosis, which is due to virus-induced increased intracellular lipid accumulation and impaired lipid release from infected cells [2]. These alterations promote viral replication that requires intracellular lipids to build up a membranous replication factory [3] and to assemble particular virions, designated lipoviroparticles. These particles have a lipid profile similar to lipoproteins (LPs) and are associated with apolipoproteins, particularly apolipoprotein E (ApoE) [4].

LPs such as very-low-density lipoprotein (VLDL) and high-density lipoprotein (HDL) are water-soluble assemblies of macromolecules comprising a lipid core of triglycerides and cholesteryl esters that is surrounded by a hydrophilic phospholipid monolayer. The latter is decorated with apolipoproteins such as ApoB and ApoE that stabilize the complex and provide a functional identity [5]. ApoE is synthesized primarily in hepatocytes and several non-hepatic tissues, including the brain, artery walls, spleen, kidney, muscle, and adipose tissue, but most LP subclasses in the plasma associate with hepatocyte-derived ApoE [5–7]. ApoE regulates the clearance of cholesterol-rich LPs from circulatory systems via its binding to receptors on the surface of hepatocytes, including heparan sulfate proteoglycan (HSPG) and low-density lipoprotein receptor (LDLR) [5]. It was reported that liver-generated ApoE is superior to ApoE from other tissues in the clearance of LP remnants [8]. Abnormal function of ApoE was found in patients with type III hyperlipoproteinemia, which is a disorder characterized by high blood levels of triglycerides and cholesterol [9,10]. Moreover, a recent study reported that liver-generated ApoE affects integrity of the brain [11]. At least 18 diseases, including Alzheimer's and cardiovascular diseases are strongly associated with *APOE* genotypes [12]. Moreover, *APOE* genotypes appear to correlate with the outcome of some viral diseases such as coronavirus disease 2019 (COVID-19) [13–16]. Notably, hepatic ApoE is an essential integral component of HCV and hepatitis B virus (HBV) particles and has been suggested to be a promising target for the development of effective HCV vaccines [17,18]. In addition, recent findings indicate that plasma ApoE and other apolipoproteins form a protein coat around secreted extracellular vesicles (EVs) and affect EV signaling function [19–21].

Despite a long history of intensive research, the trafficking, egress, and transmission route of hepatic ApoE-LPs are poorly understood. ApoE is a 299 amino acids (aa) long protein that contains an 18 aa N-terminal signal peptide targeting the protein co-translationally into the ER lumen [22,23]. ER-luminal ApoE is transported to the Golgi where it is modified by O-linked sialylation [24,25] and associates with nascent LPs containing ApoB100 and triglycerides. Thereafter, ApoE-ApoB100-containing LPs are further lipidated giving rise to mature LPs that have lower buoyant density [5,24–26]. To be secreted, mature LPs must be transported from the *trans*-Golgi network (TGN) to the plasma membrane (PM), but this process is poorly understood. By using an *in vitro* assay, Hossain and colleagues reported a novel transport vesicle delivering VLDL to the PM of rat hepatocytes, but the identity of this vesicle class is unknown [27]. At least in macrophages, secretion of ApoE follows the microtubule network along a protein kinase A and calcium-dependent pathway [28]. In addition, in a pigment cell type, ApoE associates with intraluminal vesicles (ILVs) within endosomes and is released with these vesicles in the form of exosomes [29]. The observation that inhibition of ApoE sorting to endosomes retains ApoE at the Golgi compartment argues for Golgi–endosome transport of ApoE [29]. Interestingly, a recent study reported a direct link between a distinct *ApoE* genotype and dysregulation of the endosomal-lysosomal compartment in an in vivo mouse model, arguing for an important role of this trafficking route in cellular homeostasis [30].

The endosomal compartment is also required for the export of HCV particles that are thought to follow a noncanonical secretory route [31]. Since HCV particles associate intracellularly with hepatic ApoE [16,32], hepatocyte-derived ApoE-LPs might also be released via an endosomal egress pathway. Consistently, HCV hijacks the endosomal pathway for the transmission of viral RNA genomes via endosome-derived CD63-positive EVs [33–36].

To study the egress pathway of ApoE-LPs, we established a fully functional fluorescently tagged ApoE and show that ApoE-LPs enrich in CD63-positive endosomes of hepatocytes. Intracellular ApoE-LPs and CD63-positive EV precursors associate with each other, with a fraction of them appearing to be co-secreted and intercellularly transmitted. Expanding these observations to HCV, we report that the viral replicase factor nonstructural protein 5A (NS5A) associates with ApoE. This interaction is required for the efficient release of ApoE-associated CD63-positive EVs containing viral RNA and being taken up by non-infected bystander cells. Thus, endosomal release of ApoE-LPs appears to be a physiological pathway that is exploited by HCV.

## Results

### Establishment of fully functional fluorescently tagged ApoE

Live-cell imaging of ApoE requires a suitable fluorescently tagged protein that retains full functionality. GFP was previously selected for ApoE labeling, but the ApoE-GFP fusion protein is prone to undesired cleavage and lacks full functionality [37]. To overcome this limitation, we probed alternative fluorescent protein (FP) tags that were fused to the C-terminus of ApoE. As target cells, we employed hepatocyte (Huh7)-derived cells (cell line Huh7-Lunet/CD81H), because they are well suitable for various imaging approaches [32] and highly permissive to HCV [38]. To avoid excessive overexpression, endogenous ApoE amount was reduced to undetectable level by stable knockdown, prior to lentiviral transduction of the cells with constructs encoding various ApoE fusion proteins (FPs) [32]. Western blot analysis revealed that in the case of all ApoE-redFPs, in addition to the full-length proteins (~58-kDa), truncated proteins (~46 kDa) were detected (S1A Fig). This truncation might be due to hydrolysis of the N-acylimine group of the DsRed-like chromophores in these FPs, especially under the acidic conditions in late endosomes where ApoE is expected to reside. Therefore, we tagged ApoE

with mTurquoise2 (mT2) and eYFP. Consistent with our assumption, ApoE$^{mT2}$ and ApoE$^{eYFP}$ were not fragmented (S1A Fig, upper right). Because mT2 is a rapidly-maturing cyan monomer with very low acid sensitivity (pKa = 3.1) [39], we selected this tagged ApoE for functional validation.

ApoE$^{mT2}$ was efficiently secreted into the cell culture supernatant (Fig 1A). Moreover, the association of secreted LPs with ApoE$^{mT2}$ was well comparable to the one with wildtype (wt) ApoE as determined by separation of LPs using sucrose density gradient centrifugation (peak density of ApoE$^{mT2}$ and ApoE$^{wt}$ = 1.05 vs. 1.04 g/ml, respectively) (Fig 1B). Moreover, in addition to a weak and diffuse ER-like pattern, ApoE$^{mT2}$ formed strong and dotted puncta characteristic for LPs and colocalized with ApoB, a well-established LP marker (Fig 1C). Of note, ApoE puncta were more easily detectable when visualized via mTurquoise2, particularly under live-cell imaging conditions, as compared to indirect immunofluorescence (Fig 1C). We further investigated ApoE$^{mT2}$ subcellular distribution in non-hepatic cells having undetectable levels of ApoE such as HEK293T and Hela cells [29,40]. Upon ectopic expression of ApoE$^{mT2}$, we observed a dot-like pattern in both cell lines, which was well comparable to the one detected in Huh7-Lunet cells (S1B Fig).

Next, we validated the functionality of ApoE$^{mT2}$ by probing its capacity to rescue the production of infectious HCV, which was used as readout because this virus incorporates ApoE into virions intracellularly to increase viral infectivity [32,41].To facilitate the analysis, we employed the HCV reporter virus JcR2a encoding Renilla luciferase [42]. JcR2a *in vitro* transcripts were transfected into ApoE knock-down Huh7-Lunet cells expressing ApoE$^{mT2}$ or ApoE$^{wt}$ (S1C Fig) or containing the empty expression vector. RNA replication, determined by luciferase assay and intracellular accumulation of core protein, was comparable among all 3 cell pools (S1D and S1E Fig). As expected, ApoE$^{wt}$ and ApoE$^{mT2}$ expression significantly alleviated the secretion of HCV virions as determined by quantifying extracellular HCV core protein and infectivity assay (Fig 1D). Baseline production of HCV in empty vector-transduced Huh7-Lunet cells was further reduced when we used the non-hepatic cell line HEK293T-miR122, which does not express endogenous apolipoproteins but supports HCV RNA replication [40], arguing that the expression of non-ApoE LPs in Huh7-derived cells compensates, at least in part, for ApoE deficiency [43,44] (S2 Fig). Also in these cells, ApoE$^{wt}$ and ApoE$^{mT2}$ rescued infectious HCV particle production (S2 Fig). Taken together, our data show that mT2 is a novel and well-applicable tag for labeling and functional analyses of ApoE.

## Endosomal trafficking and egress of ApoE in hepatocytes

Having established a suitable FP-tagged ApoE, we employed light and electron microscopy methods to study the trafficking and egress route of ApoE in hepatocytes. First, we confirmed the conventional trafficking route of ApoE, which starts at the ER where it is co-translationally delivered into the lumen to enter the secretory pathway [22,23]. Consistently, in Huh7-Lunet/ApoE$^{mT2}$ cells we detected reticular ApoE$^{mT2}$ signals overlapping with the ER marker PDI (Fig 2A, top row). In addition, we observed condensed ApoE$^{mT2}$ puncta in the Golgi area containing GM130, a marker of the Golgi apparatus, consistent with the assembly of ApoE-LPs at this site (Fig 2A, middle row).

To determine whether hepatocyte-derived ApoE-LPs are released via an endosomal egress pathway, we initially determined ApoE colocalization with CD63, the commonly used marker of ILVs that are sorted into late endosomes [45,46]. We detected numerous ApoE$^{mT2}$-containing structures in Golgi-devoid areas and these signals predominantly overlapped with CD63, indicating accumulation of ApoE in late endosomes (Fig 2A, bottom row). Consistently, a fraction of ApoE signals overlapped strongly with Rab7 (a marker of late endosomes), but rarely

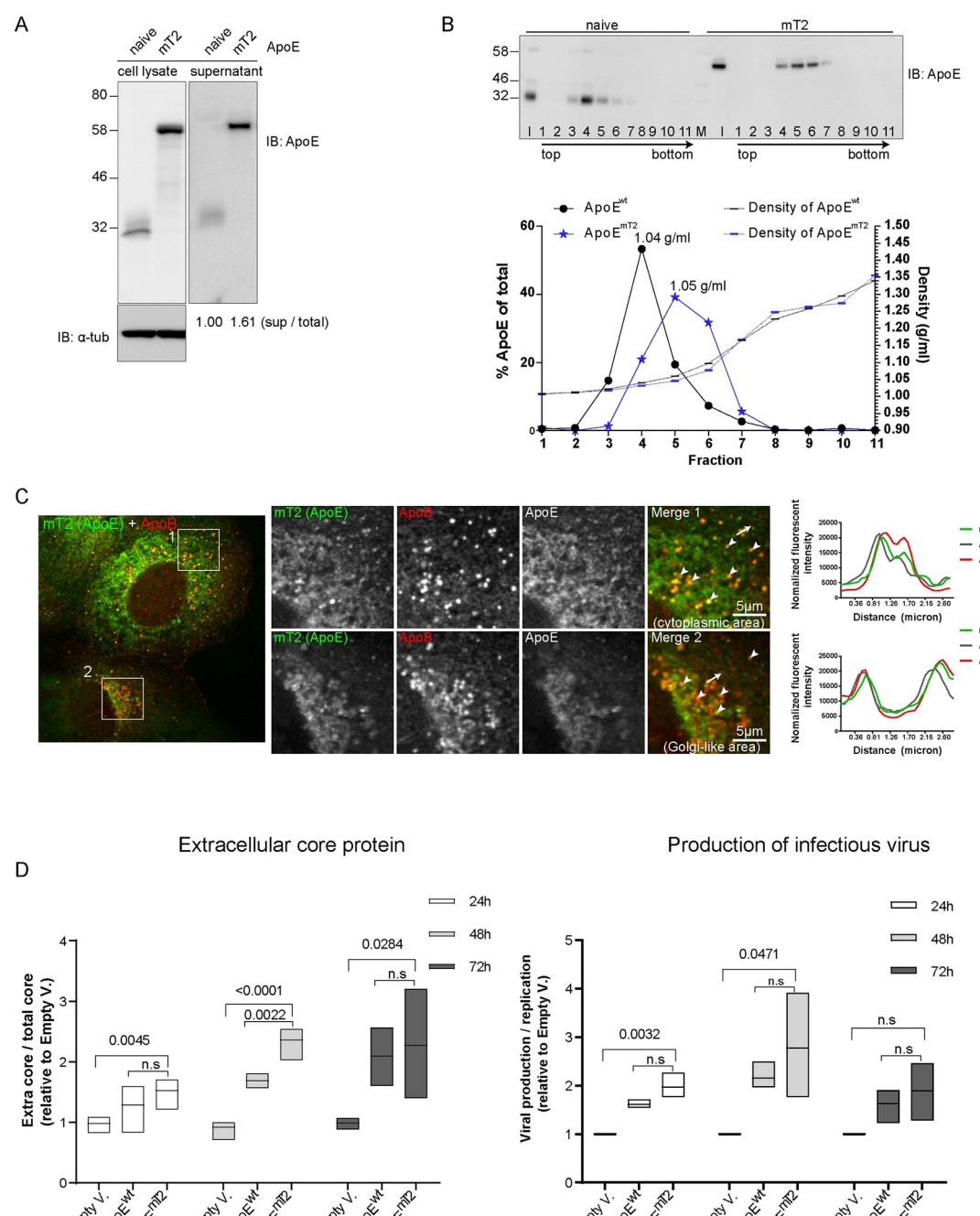

**Fig 1. Establishment and characterization of fully functional fluorescently tagged ApoE.** (A) Secretion of ApoE$^{mT2}$. Lysates and supernatants of unmodified Huh7-Lunet cells (naïve) and Huh7-Lunet cells in which endogenous ApoE was depleted by stable knock-down and substituted by stable expression of mTurquoise2-tagged ApoE (mT2) were harvested one day after seeding. Samples were analyzed by Western blot using ApoE-specific antibody; α-tubulin served as a loading control for cell lysates. The ratios of secreted to total ApoE are given below the lanes. The value of ApoE$^{wt}$ was set to 1. (B) Density of secreted ApoE$^{mT2}$. Upper panel: conditioned media of naïve Huh7-Lunet cells and Huh7-Lunet cells expressing ApoE$^{mT2}$, both from (A), were subjected to 10–50% iodixanol isopycnic centrifugation and fractions were analyzed by Western blot using ApoE-specific antibody. I: input; M: protein marker lane. Lower panel: signal intensities of the Western blot image were quantified and values were normalized to total ApoE amount in all fractions. Densities of fractions are specified on the right Y-axis (g/ml). Densities of peak fractions are given. (C) Normal lipid-binding property of ApoE$^{mT2}$. Immunofluorescent staining of ApoE$^{mT2}$ in ApoE$^{mT2}$ reconstituted Huh7-Lunet/ApoE-KD cells using ApoE- and ApoB-specific antibodies. Two-row images on the right show magnified views of boxed areas in the left overview image. Arrowheads in cropped images point to signal overlaps of ApoE$^{mT2}$ and ApoB; plot profiles in the right panels are along the lines indicated with white arrows in the merge images. (D)

Functionality of ApoE^mT2 as determined by the rescue of infectious HCV particle production. Left panel: Huh7-Lunet/ApoE-KD cells were transduced with either an empty vector (Empty V.), or ApoE^wt, or ApoE^mT2 and stably selected. Cells were then electroporated with *in vitro* transcripts of the Renilla luciferase (RLU) HCV reporter genome JcR2a. At 24, 48 and 72 h post-electroporation, amounts of extracellular core protein present in culture supernatants were determined by chemiluminescence assay. Right panel: amount of infectious HCV particles released into the culture supernatant of electroporated cells. At the indicated time points supernatants were harvested, naïve Huh7.5 cells were inoculated and 72 h later, luciferase activity was determined. Values were normalized to HCV RNA replication in each cell line to exclude replication effects. Data are medians (range) from three independent experiments. P-value was determined using unpaired Student's *t*-test. N.s: not statistically significant (P>0.05).

with ADRP (a marker of lipid droplets) (S3 Fig). Evidence for localization of ApoE to the secretory and the endosomal pathway was also obtained with the non-hepatic cell lines HEK293T and Hela that support HCV particle production upon ectopic ApoE expression [40]. When we expressed ApoE^mT2 in these cells, we observed its colocalization with both endosomal (PDI, GM130) and endocytic (CD63, Rab7) markers (S4 Fig), suggesting that also in these non-hepatic cells ApoE traffics via an endosomal pathway.

Given the important role of autophagy in the degradation of intracellular content, including material in late endosomes, we wondered whether ApoE resides in CD63-positive late endosomes of hepatocytes as result of autophagy. However, only a minor fraction of ApoE^mT2 colocalized with LC3 puncta suggesting that autophagy is not required for the enrichment of ApoE in late endosomes (Fig 2B).

To exclude the possibility that detected ApoE foci in late endosomes corresponded to secreted ApoE that was re-internalized into cells by endocytosis, we compared the number of endocytosed ApoE-containing structures, relative to total intracellular, newly synthesized ApoE structures. To this end, we cocultured freshly transduced ApoE^mT2 cells (donor) with excess number of bystander (recipient) cells expressing the human HRAS-derived CaaX peptide that was fused to eYFP for 12 h (Fig 2C, upper left panel). The CaaX motif, which is a farnesylation signal from the human HRAS protein, targets the eYFP fusion protein to cellular membranes making them easily trackable via eYFP and allowing the faithful discrimination of recipient and donor cells. In this setting, donor cell-transmitted ApoE detected in recipient cells corresponds to secreted and re-internalized ApoE. Using this approach, we observed in donor cells newly expressed ApoE forming numerous dotted structures colocalizing with about a quarter of total detected CD63 signals in these cells (Figs 2C, panel i. and S5A). Importantly, the number of re-internalized ApoE detected in CaaX^eYYFP recipient cells was very low (Fig 2C, panel i.). As second approach, we generated an expression construct encoding a functional SNAP-tagged ApoE^SNAPf (S5B Fig). In this case, cells expressing ApoE^SNAPf were cultured in medium containing the cell-nonpermeable SNAPf fluorophore for 12 h to label secreted ApoE and track its presence in cells after re-internalization (Fig 2C, lower left panel). Also in this setting, re-internalization of ApoE was negligible compared to the overall high number of ApoE-CD63 double-positive foci (panel ii. in Figs 2C and S5C). These results suggested that newly synthesized ApoE-CD63 co-labeled structures corresponded to ApoE targeted to CD63-positve endosomes in donor cells, but not to ApoE re-internalized from the extracellular medium.

We further identified the ultrastructure of ApoE-CD63 positive signals by correlative light and electron microscopy (CLEM) using lipid droplets as fiducial markers, because they are easy to detect in both light and electron microcopy and have a unique distribution and size in each Huh7 cell (Fig 2D). We found that ApoE-CD63 double-positive signals predominantly corresponded to regions containing electron-dense vesicles of ~500 nm in diameter, which is a typical feature of endosomal compartments [47] (Fig 2D, right panel).

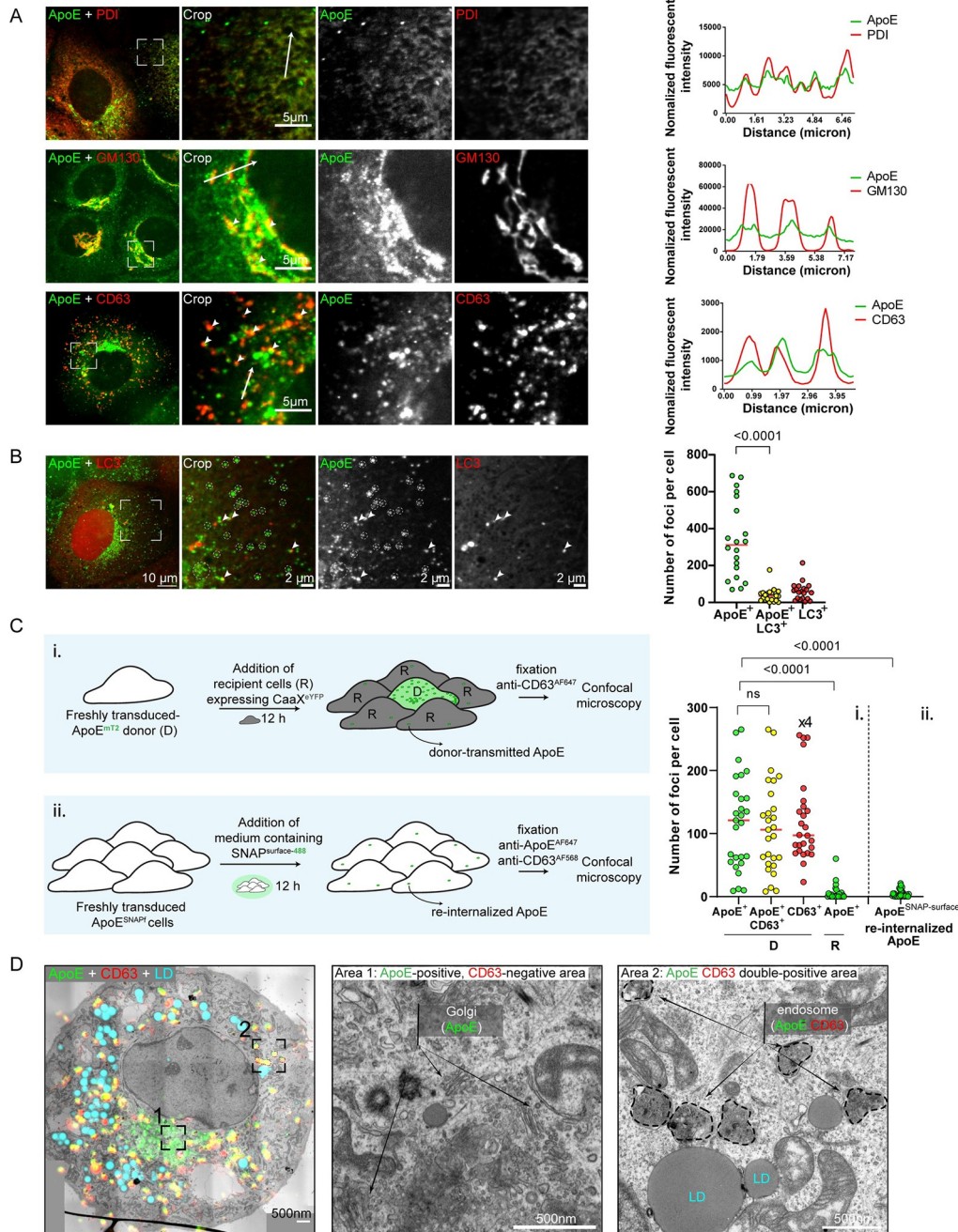

**Fig 2. Detection of ApoE in CD63-positive late endosomes, intracellular endosomal trafficking of ApoE and egress from hepatocytes.** (A) Colocalization of ApoE$^{mT2}$ with markers of the ER (PDI), Golgi (GM130), and intraluminal vesicles/endosomes (CD63). Proteins specified on the top of each panel were detected in Huh7-Lunet/ApoE$^{mT2}$ cells by immunostaining and cells were analyzed by confocal microscopy. Boxed areas in the left panels are shown as enlarged views in the panels on the right of each row. Profiles on the right of each panel were taken along the lines indicated with white arrows in cropped images. (B-C) ApoE foci predominantly form independent of autophagy and re-internalization of secreted ApoE via endocytosis. (B) Majority of ApoE$^{mT2}$ foci is devoid of LC3 puncta. Huh7-Lunet/ApoE$^{mT2}$ cells were lentivirally transduced with the LC3$^{mCherry}$ expression construct. At 24 h post-transduction, cells were fixed and analyzed by confocal microscopy. Boxed area in the left panel is shown as enlarged view in the panels on the right. Dashed circles mark LC3-free ApoE$^{mT2}$ foci. Arrowheads point to LC3-positive ApoE$^{mT2}$ signals. The numbers of total ApoE, LC3, and ApoE-LC3 double-positive foci in single cells were determined (right). P-values were determined using Mann-Whitney test. (C) Majority of ApoE-CD63 double-positive signals corresponds to newly synthesized structures rather than secreted and re-internalized structures. Schematic representation of used approach to visualize newly synthesized and re-internalized ApoE. (i) Huh7-Lunet/ApoE-KD cells were transduced with lentiviruses encoding

ApoE[mT2] for 4 h (donor/D). After being washed twice with PBS, the cells were added to Huh7-Lunet/ApoE-KD cells expressing the membrane marker eYFP[CaaX] (recipient/R) at a 1:4 (D:R) ratio. After 12 h, cells were fixed, subjected to immunostaining for CD63, and analyzed by confocal microscopy. (ii) Huh7-Lunet/ApoE-KD cells, lentivirally transduced with the ApoE[SNAPf] expression construct for 4 h, were cultured in medium containing 5 μM cell-nonpermeable SNAP-surface substrate for 12 h. Thereafter, cells were fixed, subjected to ApoE and CD63 immunostaining, and analyzed by confocal microscopy. Right panel: quantitative analysis of the two assays. (i) Number of total ApoE, CD63, and ApoE-CD63 double-positive foci in single donor cells (D) and donor-transmitted ApoE-positive foci in single recipient cells (R). Values of total CD63[+] signals were divided by 4 to account for their high abundance; (ii) number of re-internalized ApoE foci in single cells. P-values in both approaches were determined using Mann-Whitney test. (D) Endosomal localization of ApoE-CD63 double-positive structures. Huh7-Lunet/ApoE[mT2] cells expressing CD63[mCherry] were analyzed by CLEM using lipid droplets (LDs) stained with lipidTox as fiducial markers. The overlay image is shown on the left. Middle and right panels: magnified EM micrographs from an area with ApoE-positive, CD63-negative signals showing Golgi stacks and vesicles (crop 1) and from an area with ApoE-CD63 double-positive endosomes (crop 2), respectively.

With the aim to track and record the dynamics of ApoE association with CD63, we took advantage of ApoE[mT2] and conducted time-lapse confocal microscopy (S1 Movie). Initially, we observed co-trafficking of ApoE-CD63 double-positive puncta as indicated by their similar mean squared displacement values (Fig 3A, top lines). Particle size and velocity of ApoE-CD63 double-positive signals were also computed, revealing substantial heterogeneity of particle motions (S6A Fig, left and right). Importantly, a subpopulation of these vesicles displayed directed motions (S6B Fig, left), suggesting microtubule-dependent trafficking of late endosomes containing ApoE and CD63 [48,49]. An example of ApoE-CD63 co-trafficking dynamics showing a directed motion towards the cell periphery is shown in S6B Fig, right panel.

To further dissect the possible involvement of the microtubule network in the trafficking of ApoE-CD63-positive endosomes in hepatocytes, we treated Huh7-Lunet/ApoE[mT2]/CD63[mCherry] cells with colchicine to depolymerize microtubules [28]. As expected, cells showed significant abrogation of ApoE and CD63 motility as deduced by their smaller MSD values over time (dotted lines in Fig 3A and S2 Movie). In addition, quantitative analysis revealed no major differences between the sizes of ApoE-CD63 double-positive particles in mock and colchicine-treated conditions (Fig 3B), but their velocities were dramatically reduced in the latter case (Fig 3C). Interestingly, in mock-treated condition, we frequently observed a large population of ApoE-CD63 double-positive particles at the cell periphery (Fig 3D, arrows in upper panels), but this population was profoundly reduced upon colchicine treatment (Fig 3D, lower panels). Quantitatively, radial CD63 signal intensity in the cell periphery of colchicine-treated cells decreased significantly (Fig 3E), suggesting that the intact microtubule network is required for the transport of both total CD63[+] and ApoE-containing CD63[+] endosomes to the cell periphery. Consistently, Huh7-Lunet cells treated with various concentrations of colchicine (5–80 μM) for 1 h released significantly less ApoE into the cell culture supernatant as determined by Western blot (S6C Fig). In addition, we employed Nano-luciferase (Nluc)-tagged CD63 (CD63-Nluc)-expressing cells to allow the rapid and sensitive measurement of CD63 secretion [50–52], along with the quantification of ApoE secretion in mock-and colchicine-treated cells by Elisa. We observed a parallel reduction of ApoE and CD63 secretion (Fig 3F and 3G). Taken together, our data argue for a proportion of intracellular ApoE egressing via the CD63 endosomal pathway in a microtubule-dependent manner.

To visualize the intracellular trafficking of ApoE-associated CD63-positive ILVs, we took advantage of the acidic pH in endosomes that gets neutral as endosomes fuse with the PM to release ILVs contained therein. As an endosome-PM fusion sensor, we employed an improved version of pHluorin [53] that was inserted into the first extracellular loop of CD63, thus exposing pHluorin to the acidic environment of the endosomes. The signal of pHluorin-tagged CD63 is quenched in the endosomes and exclusively fluoresces upon exposure of the

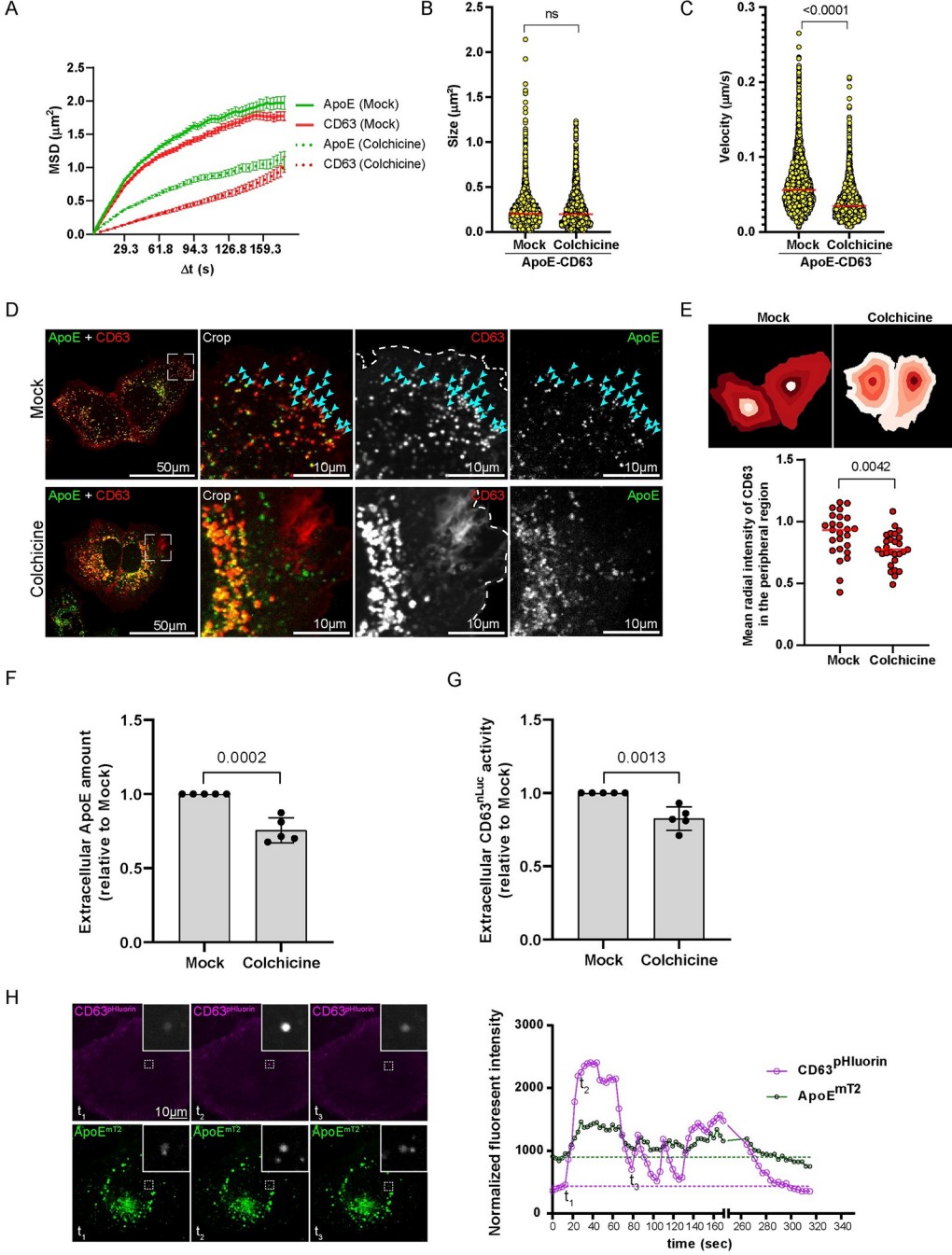

**Fig 3. Endosome-dependent trafficking and secretion of ApoE.** (A-C) Microtubule-dependent motility of intracellular ApoE-CD63 double-positive structures. (A) Mean squared displacement (MSD) of general ApoE and CD63 trafficking in mock and colchicine-treated cells. Huh7-Lunet/ApoE$^{mT2}$ cells expressing CD63$^{mcherry}$ were either mock-treated or treated with 80 μM colchicine for 1 h to depolymerize microtubules and analyzed by live-cell confocal microscopy. (B) Sizes of ApoE-CD63 double-positive structures from (A) were measured. P-value was determined using Mann-Whitney test. Ns, non-significant. (C) Trafficking velocities of ApoE-CD63 double-positive structures were computed (see methods for detail). P-value was determined using Mann-Whitney test. (D-E) Peripheral distribution of ApoE-CD63 double-positive structures depends on intact microtubules. (D) Representative images of cells analyzed by live-cell confocal microscopy from (A). Boxed areas in the left panels are shown as enlarged views in the panels on the right of each row. Arrowheads point to peripheral ApoE-CD63 double-positive signals in mock-treated cells. (E) The mean radial signal intensities of CD63 in the peripheral regions of cells from (D) were determined (top) and quantified (bottom) by using the CellProfiler module "MeasureObjectRadialDistribution". P-value was determined using Mann-Whitney test. (F-H) Secretion of ApoE via the endosome-dependent route. (F) Huh7-Lunet cells expressing CD63$^{nLuc}$

were either mock-treated or treated with 5 μM colchicine for 1 h to depolymerize microtubules. Extracellular ApoE in the supernatant was quantified by ApoE Elisa. Data are means (SD) from 5 independent experiments. P-value was determined using unpaired Student's *t*-test. (G) The extracellular CD63$^{\text{nLuc}}$ activity in the supernatants from (F) was quantified by nLuc assay. Data are means (SD) from 5 independent experiments. P-value was determined using unpaired Student's *t*-test. (H) Secretion of ApoE-positive ILVs visualized by pHluorin-tagged CD63. Huh7-Lunet cells expressing ApoE$^{\text{mT2}}$ and CD63$^{\text{pHluorin}}$ were cultured in imaging medium (pH 7.4) and analyzed by time-lapse confocal microscopy with a focus on plasma membrane resident CD63-fluorescent signals. (Right panel) Maximum fluorescence intensity of CD63$^{\text{pHluorin}}$ and associated ApoE in the selected dashed area indicated in S3 Movie. Images taken at indicated time points are displayed on the bottom and they correspond to initiation (t1), peak (t2), and late-secretion (t3), respectively.

endosomes' interior to the neutral pH of the extracellular environment [54]. To capture ApoE-CD63 co-secretion, we used time-lapse live-cell confocal microscopy by setting the focal plane to the PM as determined by the basal fluorescence of the CD63$^{\text{pHluorin}}$ signal (Fig 3H and S3 Movie). As expected, CD63$^{\text{pHluorin}}$ expressed in Huh7-Lunet cells showed a predominant fluorescent signal in the PM. Of note, we observed occasional steep and rapid increases of the vesicular ApoE-associated pHluorin signal (Fig 3H, time point t$_2$) corresponding most likely to the fusion of ApoE-CD63 containing endosomes with the PM and thus, the release of ApoE-associated CD63-positive ILVs.

## Association of ApoE-containing lipoproteins with CD63-positive extracellular vesicles and their possible intercellular transmission

A recent study by Busatto and colleagues demonstrated that EVs in crude plasma frequently bind to and fuse with LPs arguing for a physiological interaction between these two particle species [19]. Given the cotrafficking of intracellular hepatic ApoE with CD63 (Fig 3), we speculated that extracellular hepatic ApoE might associate with CD63-positive EVs via LPs. Given the difficulties to separate EVs from LPs [20,55–57], we employed ApoE-specific pull-down to isolate ApoE from the supernatant of Huh7-Lunet cells that had been cultured in EV-depleted medium. Captured complexes were eluted under native conditions and analyzed by EM revealing predominantly small vesicles, which had the size of regular LDL or large HDL particles (mean diameter ~25 nm) (Fig 4A). Of note, we detected in much lower quantity co-captured bigger vesicles (mean diameter ≥50 nm) (Fig 4A, labeled with stars), a fraction of them staining positive for CD63 and being associated with the smaller ApoE-positive particles (Fig 4B). This result argued for the stable interaction between secreted ApoE-LPs and CD63-positive EVs, consistent with a previous study [19].

Next, we examined the cell-to-cell transfer of both ApoE-LPs and CD63-positive EVs. To this end, we used hepatic donor cells expressing fluorescently labeled ApoE$^{\text{mT2}}$ and CD63$^{\text{mCherry}}$, and recipient cells expressing CaaX$^{\text{eYFP}}$ (Fig 4C and 4D, gray cells). Cells were seeded into imaging dishes and 16 h later, examined by live-cell confocal microscopy. We observed both donor-derived ApoE$^{\text{mT2}}$ single-positive and ApoE$^{\text{mT2}}$-CD63$^{\text{mCherry}}$ double-positive structures in recipient cells, indicating transfer and uptake of both structures (Fig 4D and S4 Movie). Quantitative image analysis revealed a time-dependent increase in the number of ApoE$^{\text{mT2}}$-CD63$^{\text{mCherry}}$ double-positive structures in single recipient cells, especially at 48 h post-seeding (Fig 4E). Taken together, these results suggest that hepatic ApoE-LPs are secreted and transferred from cell to cell on their own and possibly also in association with CD63-positive EVs.

## Intracellular enrichment of HCV NS5A in ApoE-positive structures independent from virion assembly

As alluded to in the introduction, ApoE associates with HCV particles, most likely via the viral envelope glycoprotein complex E1/E2 [32,58] and with the viral replicase factor NS5A

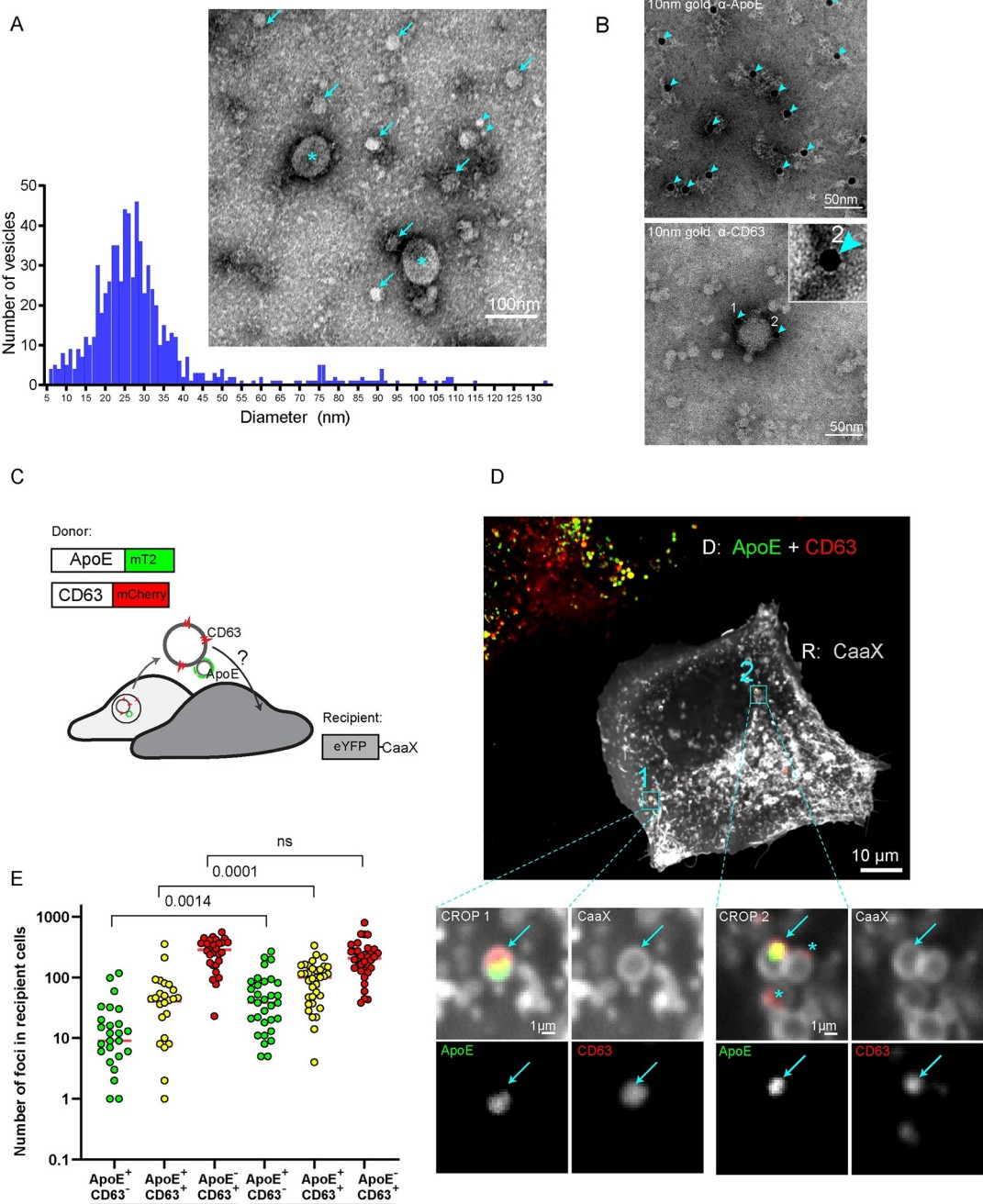

**Fig 4. Association of ApoE-containing lipoproteins with CD63-positive extracellular vesicles and their endocytosis.** (A) Analysis of fractions enriched for ApoE-containing LPs. Huh7-Lunet cells were cultured in EV-depleted medium and ApoE-associated vesicles released into the culture medium were captured using ApoE-specific antibody. Immunocomplexes were analyzed by TEM after negative staining. Arrowheads: ~5–10 nm vesicles; arrows: ~20–30 nm vesicles; stars: ~50–60 nm vesicles. Vesicles in the electron micrographs were segmented by using Ilastik to allow quantification of vesicle diameters shown in the histogram below the micrograph. (B) Detection of CD63-positive EVs in isolated ApoE-containing LP fractions. Purified ApoE-associated vesicles from (A) were immunogold-labeled with ApoE- (upper) and CD63-specific antibodies (lower). Arrowheads point towards gold particles. A zoom image of a CD63-positive gold particle is shown on the top. Note that immuno-gold labeling of ApoE alters the vesicular shape of LPs, most likely because of distortion of ApoE by used antibodies and protein A-gold incubation during the labeling procedure known to destabilize the LP structure. (C-E) Visualization of the uptake of ApoE and CD63 by recipient cells. (C) Schematic representation of used approach. Huh7-Lunet/ApoE$^{mT2}$/CD63$^{mCherry}$ served as donor cells; Huh7-Lunet cells expressing eYFP-tagged CaaX (the farnesylation signal from human HRAS) as recipients. (D) Donor and recipient cells from (C) were co-cultured for 16 h and analyzed by live-cell

confocal imaging (refers to S4 Movie). D: donor; R: recipient. Arrows in cropped sections on the bottom indicate transferred ApoE-CD63 signals; stars: transferred CD63-only signals. (E) Donor and recipient cells from (C) were co-cultured and fixed at 24 h and 48 h post-seeding. The numbers of ApoE$^+$ CD63$^-$ and ApoE$^-$ CD63$^+$ single positive structures as well as ApoE$^+$ CD63$^+$ double-positive structures in single recipient cells were quantified. Each dot represents a single cell. P-value was determined using Mann-Whitney test.

[59–61]. While the ApoE-E1/E2 interaction appears to be critical for HCV particle production, NS5A has been detected in purified EV preparations [62,63], raising the question of whether NS5A follows the ApoE endosomal egress pathway. To address this question, we monitored ApoE, NS5A and E2 trafficking in HCV-replicating Huh7-Lunet cells stably expressing ApoE$^{mT2}$. FPs for NS5A and E2 were chosen to allow clear spectral separation from each other and from ApoE$^{mT2}$. In each case, fusion with the FP did not affect the functionality of the protein as shown here for ApoE$^{mT2}$, and earlier for tagged NS5A and E2 [64,65]. To allow live-cell imaging under low biosafety conditions, we took advantage of the HCV trans-complementation system [66] in which the HCV genome is genetically split into a stably expressed unit encoding the viral assembly factors (core-E1-E2$^{eYFP}$-p7-NS2) and a self-replicating subgenomic replicon encoding the viral replicase proteins (NS3-4A-4B-5A$^{mCherry}$-5B) (Fig 5A). To determine the overall subcellular distribution of FP-tagged ApoE$^{mT2}$, NS5A$^{mCherry}$, and E2$^{eYFP}$ during the course of HCV infection, we acquired time-lapse images by confocal spinning-disk microscopy in 30 min intervals between 5 and 54 h post-electroporation using minimum laser exposure to avoid phototoxicity. Prior to electroporation of the subgenomic replicon, E2$^{eYFP}$ showed a reticular ER-like pattern consistent with its ER retention [67]. Around 25 h post-electroporation, E2$^{eYFP}$ subcellular distribution began to change and NS5A$^{mCherry}$-E2$^{eYFP}$ double-positive foci became visible (arrowheads in Fig 5B and S5 Movie) [65]. In addition, ApoE$^{mT2}$-NS5A$^{mCherry}$-E2$^{eYFP}$ triple-positive foci, putative sites of HCV assembly, were observed, but their abundance was very low (Fig 5B, stars). Consistent with ongoing HCV replication, NS5A$^{mCherry}$ signal intensity increased steadily and NS5A$^{mCherry}$-ApoE$^{mT2}$ double-positive foci formed. Their abundance increased significantly over time (Fig 5C), much higher as compared to NS5A$^{mCherry}$-E2$^{eYFP}$ positive foci. We confirmed the high number of NS5A$^{mCherry}$-ApoE$^{mT2}$ double-positive foci at a late stage of infection by live-cell imaging using a shorter time interval (1 frame every 10 s). Under this imaging condition, NS5A$^{mCherry}$-ApoE$^{mT2}$ foci were readily detectable (S6 Movie).

To confirm the formation of NS5A-ApoE double-positive structures in the context of a full-length HCV genome, we transfected Huh7-Lunet/ApoE$^{mT2}$ cells with *in vitro* transcripts of a cloned HCV genome and determined NS5A and ApoE subcellular distribution in relation to the ER marker PDI by immunofluorescence. Also under these conditions, ApoE signals significantly overlapped with NS5A, confirming that the trans-complementation system recapitulates events occurring in natural infection (S7A and S7B Fig) although efficiency of HCV assembly might be lower than in authentic infection.

Next, we determined whether formation of NS5A-ApoE positive structures depends on viral assembly or is linked to some other events such as the formation of EVs. To this end, we used the same experimental approach as shown in Fig 5A, but employed a construct lacking the viral assembly factor NS2 (core-E1-E2$^{eYFP}$-p7) (Fig 5D, upper panel) [68]. While under these conditions NS5A$^{mCherry}$-E2$^{eYFP}$ double-positive structures were no longer detected, NS5A$^{mCherry}$-ApoE$^{mT2}$ double-positive dots still formed (Fig 5D and 5E). These results suggested that enrichment of NS5A in ApoE-positive puncta does not depend on HCV assembly.

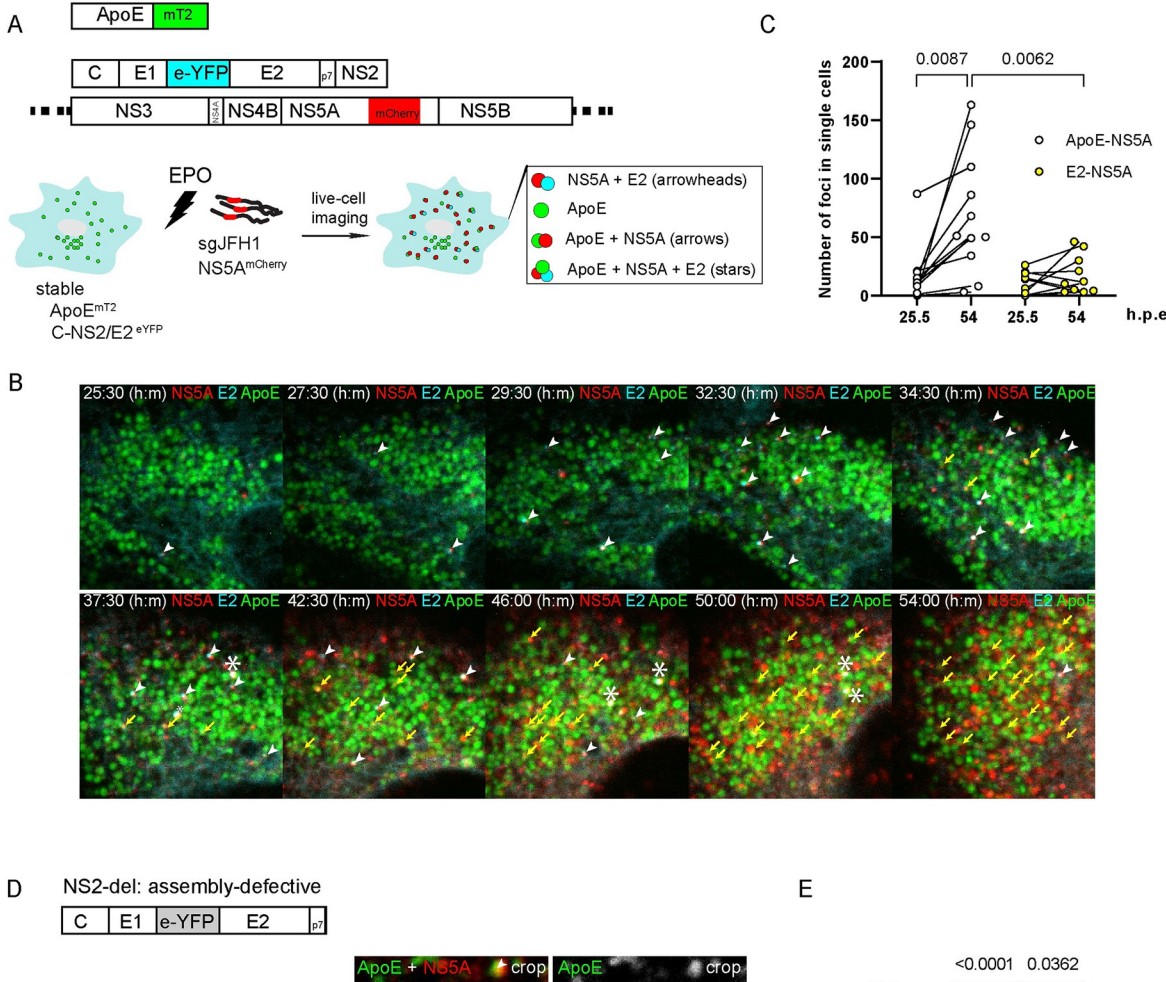

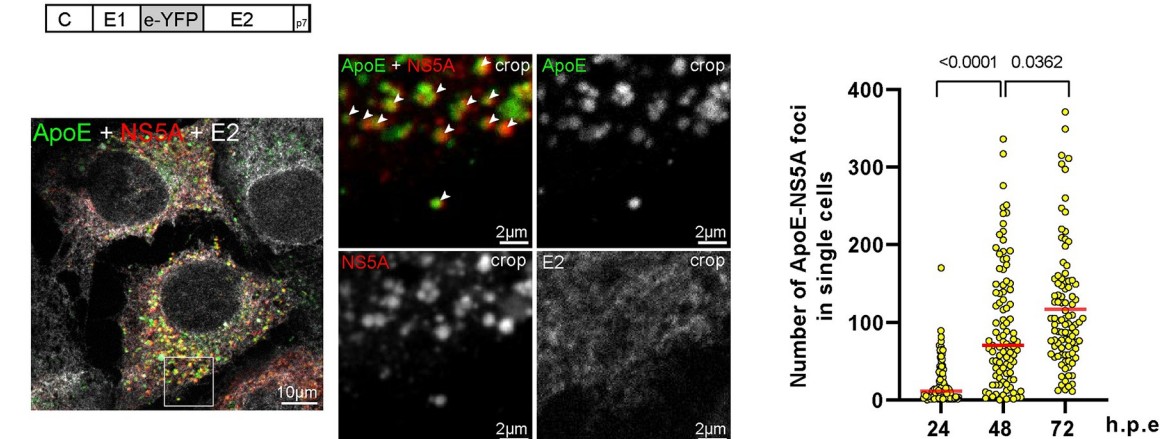

**Fig 5. Enrichment of NS5A in ApoE-positive structures and co-trafficking of ApoE$^{mT2}$ with NS5A and E2 independent of HCV assembly.** (A) Experimental approach. Fluorescently tagged ApoE$^{mT2}$, HCV proteins supporting assembly (C to NS2 with eYFP-tagged E2), and a subgenomic replicon (dotted lines indicate 5' and 3' NTRs) are shown from top to bottom; the experimental approach is depicted below. Cells stably expressing ApoE$^{mT2}$ and C-NS2/E2$^{eYFP}$ were electroporated with the replicon RNA encoding mCherry-tagged NS5A. Cells were subjected to confocal time-lapse live-cell imaging to monitor signal overlaps of the various fluorescent proteins: NS5A + E2; ApoE only; ApoE + NS5A; ApoE + NS5A + E2. (B) Time-dependent enrichment of NS5A-ApoE double-positive structures in HCV-replicating cells. Huh7-Lunet/ApoE$^{mT2}$ cells expressing HCV Core-NS2/E2$^{eYFP}$ and containing the subgenomic replicon were subjected to live-cell confocal imaging from 5 to 54 h post-electroporation (30 min/frame) to observe ApoE, NS5A, and E2 signals. A series of still images taken at time points after electroporation specified on the top are shown. White arrowheads: NS5A-E2 foci; yellow arrows: ApoE-NS5A foci; stars: ApoE-NS5A-E2 triple-positive foci. (C) Quantification of NS5A-ApoE and E2-NS5A double-positive foci detected in single cells in (B). Ten single cells were analyzed. P-value was determined using Mann-Whitney test. (D) Assembly-independent enrichment of NS5A in ApoE-positive foci. Huh7-Lunet/ApoE$^{mT2}$ cells expressing the C-p7 construct (NS2-deletion; upper panel) were electroporated with *in vitro*-transcripts of the subgenomic replicon sgJFH1/NS5A$^{mCherry}$ and analyzed by confocal microscopy to observe

ApoE, NS5A, and E2 signals. A representative image showing ApoE-NS5A double-positive foci (arrowheads) and diffuse E2 signal at 72 h post-electroporation is shown. Images on the right show magnified views of the boxed area in the left overview image. (E) Quantification of NS5A-ApoE double-positive foci detected in 100 single cells in (D) at 24, 48, and 72 h post-electroporation. Data are medians (range) of the number of detected foci. P-value was determined using Mann-Whitney test.

## Formation of NS5A- and ApoE-containing intraluminal vesicles in CD63-positive endosomes

Since HCV has been reported to transmit its RNA via a noncanonical pathway comprising endosome-derived CD63-positive EVs [34–36,69] and because ApoE-LPs also egress along the CD63-positive late endosomal pathway (Figs 2, 3 and 4), we characterized the association of ApoE-NS5A double-positive structures with CD63 in greater detail by using super-resolution microscopy. To make this possible, we replaced the FPs of ApoE$^{mT2}$ and NS5A$^{mCherry}$ by SNAPf and CLIPf, respectively (Fig 6A, upper). Both fusion proteins were fully functional as revealed by the secretion of ApoE$^{SNAPf}$ and the replication competence of NS5A$^{CLIPf}$ when tested in the context of a subgenomic replicon (S5A, S7C and S7D Figs, respectively). We further examined the labeling of these proteins in HCV-replicating cells. To this end, we transfected Huh7-Lunet/ApoE$^{SNAPf}$ cells with HCV sgRNA encoding NS5A$^{CLIPf}$ and labeled these proteins with medium containing the dyes SNAP$^{SIR647}$ and CLIP$^{ATTO590}$ for 1 h at 48 h post-electroporation. Thereafter, cells were washed to remove unbound dyes and subjected to live-cell imaging or fixed-cell microscopy (Fig 6A, lower). Confocal imaging of the cells revealed specific labeling of ApoE$^{SNAPf}$ and NS5A$^{CLIPf}$ and strong colocalization of both proteins (Fig 6B), consistent with our previous results with FP-tagged ApoE and NS5A (Fig 5). Importantly, we found that about half of ApoE—NS5A double-positive foci also contained CD63 (Figs 6C and S7E). When we visualized NS5A and ApoE by stimulated emission depletion (STED) super-resolution microscopy, in addition to the reticular ER and the ring-like lipid droplet staining patterns of NS5A$^{CLIPf}$, we detected ~100–200 nm diameter NS5A$^{CLIPf}$-positive structures that were decorated with ApoE$^{SNAPf}$ at CD63-positive sites (Fig 6D, arrows).

To determine the ultrastructure of ApoE-NS5A double-positive sites, we employed CLEM using Huh7-Lunet/ApoE$^{mT2}$ cells expressing the HCV assembly factors and containing a subgenomic replicon (refer to Fig 5A). We observed an overlap of NS5A$^{mCherry}$-ApoE$^{mT2}$ double-positive signals with endosomes (Fig 7A and 7B). Strikingly, inside these endosomes, we detected numerous ILVs with double or multi-membrane bilayers (arrowheads in crop 1, 2, and 3 of Fig 7B), which were only rarely detected in NS5A$^{mCherry}$-negative, ApoE$^{mT2}$-positive endosomes (crop 4). Sites of NS5A$^{mCherry}$-E2$^{eYFP}$ double-positive structures, putative HCV assembly sites, overlapped with HCV-induced accumulations of double-membrane vesicles (DMVs), the presumed sites of viral RNA replication that were often found in close proximity to lipid droplets (Fig 7B, crop 5 and 6) as reported earlier [65]. Taken together, these results argued for the accumulation of NS5A- and ApoE-containing ILVs at sites of CD63-positive endosomes.

To determine whether the enrichment of NS5A in ApoE-containing endosomes is autophagy-dependent, we employed autophagy-deficient Huh7-derived cells with stable knockout (KO) of ATG5 and ATG16L1 [70]. We observed a comparatively high number of ApoE foci in these KO cells and KO-control cells suggesting that autophagy is not required for the sorting of ApoE in late endosomes (S8A and S8B Fig), which is consistent with our previous observation (Fig 2B). Moreover, enrichment of NS5A in ApoE-containing endosomes is not strictly autophagy-dependent, because the numbers of ApoE-NS5A double-positive foci were comparable in autophagy-deficient and KO-control cells (S8A and S8B Fig). Interestingly, a minor fraction of these foci colocalized with LC3 puncta in KO-control cells, but not in autophagy-

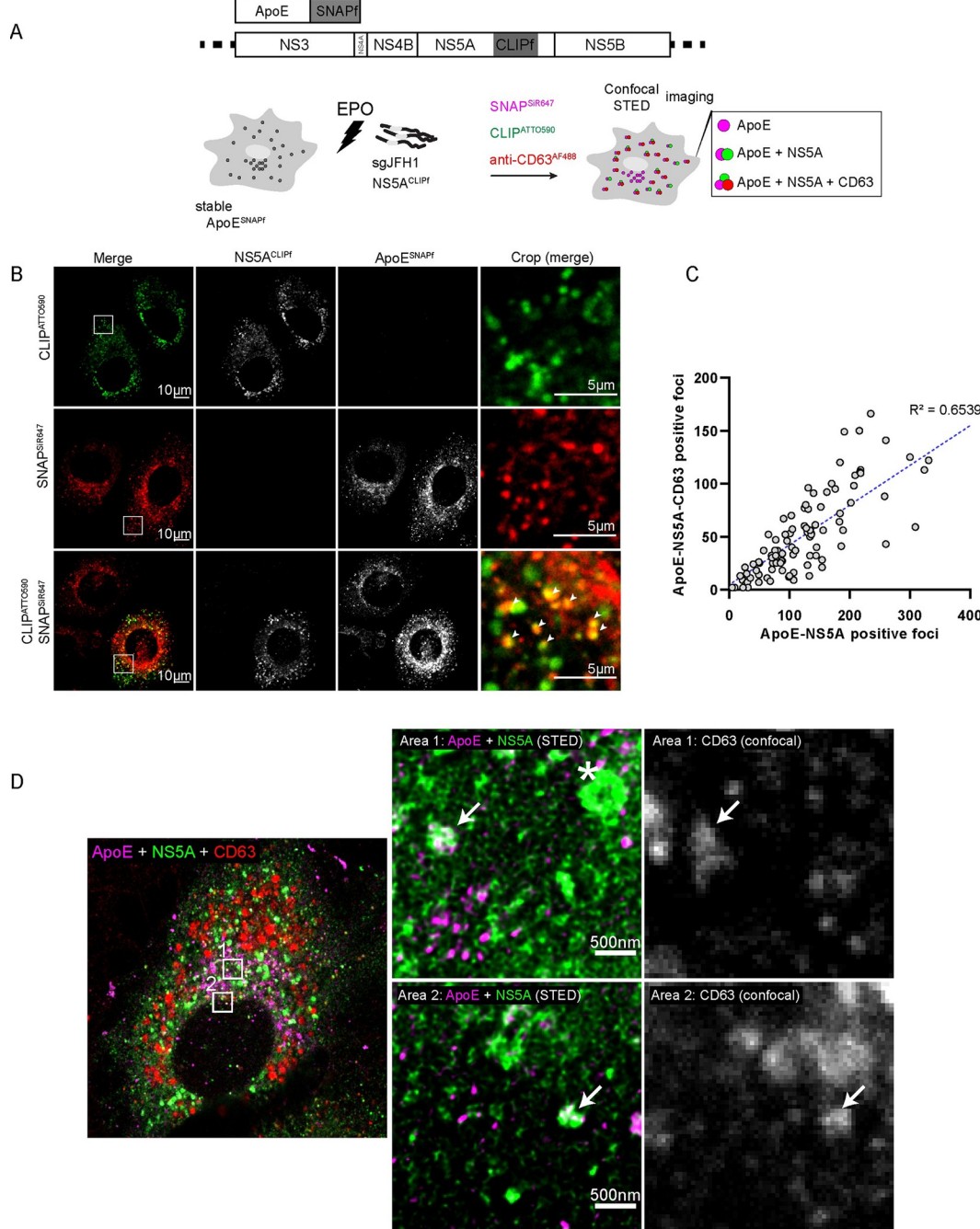

**Fig 6. Colocalization of NS5A and ApoE with the intraluminal vesicle marker CD63 as revealed by super resolution microscopy.** (A) Experimental approach. Schematic representations of SNAPf-tagged ApoE and the subgenomic replicon encoding CLIPf-tagged NS5A are shown on the top. Huh7-Lunet cells were lentivirally transduced with the ApoE expression vector and transfected with the subgenomic replicon RNA. ApoE and NS5A were detected by STED microscopy and CD63 by immunofluorescence confocal microscopy. (B) Colocalization of ApoE$^{SNAPf}$ and NS5A$^{CLIPf}$. Huh7-Lunet/ApoE$^{SNAPf}$ cells were electroporated with subgenomic replicon RNA encoding NS5A$^{CLIPf}$ and after 48 h, cells were labeled with SNAP$^{SiR647}$ and CLIP$^{ATTO590}$ for 1 h, fixed, and subjected to confocal microscopy. Arrowheads: colocalized ApoE-NS5A signals. (C) Quantification of CD63-positive ApoE-NS5A double-positive foci. Cells from (B) harvested at 72 h post-electroporation were fixed, permeabilized, and incubated with anti-CD63$^{AF488}$ antibody. To determine the correlation between ApoE-NS5A double-positive foci and how many of them colocalized with CD63, 100 cells were analyzed. Each dot represents one cell and displays the number of ApoE-NS5A double-positive foci (x-axis) and the number of CD63-ApoE-NS5A triple-positive foci (y-axis). The R-squared value is given on the plot. (D) STED-resolved ApoE-NS5A double-positive structures colocalizing with the intraluminal vesicle marker CD63. Huh7-Lunet/ApoE$^{SNAPf}$ cells were electroporated with the subgenomic replicon

RNA encoding NS5A$^{CLIPf}$. After 48 h, cells were labeled with SNAP$^{SiR647}$ and CLIP$^{ATTO590}$ for 1 h, fixed, and incubated with anti-CD63$^{AF488}$ antibody. ApoE, NS5A, and CD63 fluorescent signals were sequentially imaged using confocal and STED microscopy, the latter to achieve a higher resolution of ApoE and NS5A signals that were deconvoluted using Huygens. Arrows: ~100–200 nm-sized ApoE-NS5A-CD63 positive structures; star: ~500 nm-sized ring-like NS5A positive structure.

deficient cells (S8A, arrowheads and S8B Fig), suggesting that ApoE-NS5A double-positive endosomes might have fused with autophagosomes. Of note, we did not readily observe enrichment of NS5A in ApoE$^+$ foci in the context of single NS5A protein expression; instead, under those conditions NS5A signals appeared to localize in ApoE-devoid areas (S8C Fig).

## Association and transmission of ApoE-positive lipoproteins with extracellular vesicles containing HCV NS5A and RNA

ApoE associates with NS5A in regions of endosomes containing HCV-produced intraluminal double or multi-membrane vesicles (Figs 6 and 7). Moreover, HCV suppresses the fusion of late endosomes with lysosomes [71]. Therefore, we hypothesized that secreted ApoE-LPs might associate with HCV-produced EVs containing NS5A and viral HCV RNA. To address this assumption, we employed a subgenomic HCV replicon that supports viral RNA transmission via endosome-derived EVs, albeit with a rather low efficiency [33–36]. In the first set of experiments, we determined whether ApoE associates with NS5A and viral RNA released from cells containing a stable subgenomic HCV replicon or parental control cells by using ApoE-specific pull-down. Captured complexes were analyzed by HCV-specific RT-qPCR. As shown in Fig 8A, we detected HCV RNA in immuno-captured ApoE-containing complexes isolated from supernatants of replicon-containing cells. Samples captured with control antibodies or from mock-transfected cells were at the background level arguing for the release of EVs containing viral RNA from replicon cells.

To verify the presence of NS5A in the ApoE-captured complexes, we transfected Huh7-Lunet cells with a subgenomic replicon RNA encoding Nluc-tagged NS5A to allow its sensitive detection in cell culture supernatants (Fig 8B, upper). In agreement with a previous report [63], we observed time-dependent secretion of NS5A$^{Nluc}$ into the cell culture supernatant (S7 F Fig). Importantly, Nluc activity was clearly detected upon ApoE-specific immunocapture indicating a direct or indirect association between NS5A and ApoE (Fig 8B, lane 2). The specificity of the pull-down was confirmed by using mock cells or an unrelated antibody (Fig 8B, lane 1 and 3, respectively). Surprisingly, the highest Nluc activity was detected in NS5A-captured immunocomplexes, arguing that NS5A is well-accessible on the outside of EVs (Fig 8B, lane 4).

Aiming to determine the direct association of HCV-produced EVs and LPs, we analyzed immunocaptured samples by negative-staining and TEM. Staining of complexes captured with NS5A- or ApoE-specific antibodies revealed EV-like structures with diameters of ~100 nm that were frequently associated with LP-like particles (Fig 8C, arrows). As expected, in ApoE-specific pulldown, LPs were more frequent, yet these samples also contained EVs associated with LPs. These results show the direct association of HCV-produced EVs with LPs.

Next, we examined the possible relevance of ApoE-NS5A interaction for the secretion of EVs containing HCV RNA. To this end, we used an NS5A mutant (APK99AAA) reported to have a defect in interaction with ApoE (S7G Fig) [60]. Of note, Huh7-Lunet cells containing a stable subgenomic replicon encoding mutant NS5A$^{APK99AAA}$ released much lower amounts of HCV-RNA than the wildtype replicon (Fig 8D). These results suggest that ApoE-NS5A interaction is required for the efficient release of EVs containing viral RNA, providing an explanation for the association of ApoE with NS5A-positive EVs released from HCV-replicating cells.

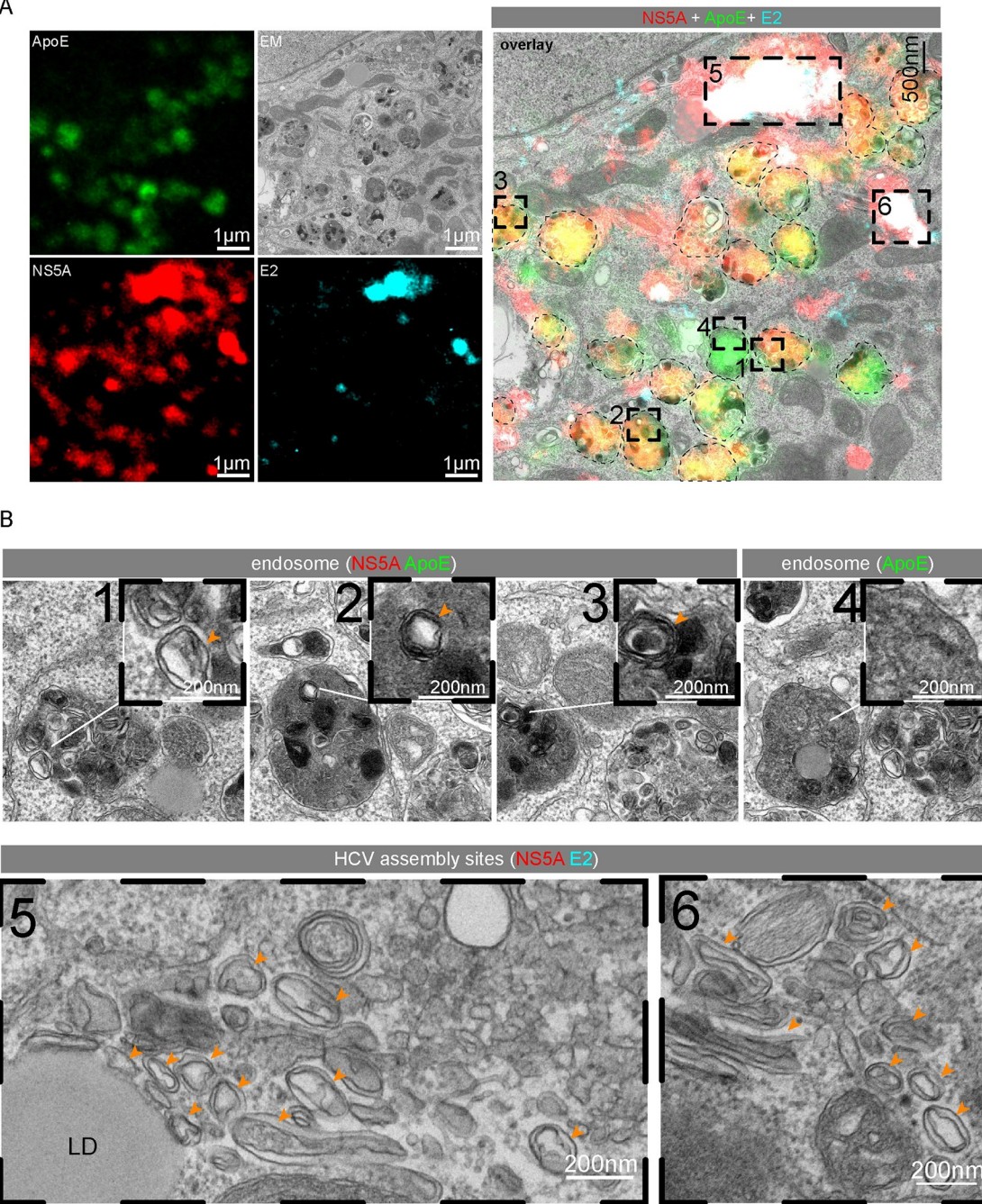

**Fig 7. Detection of HCV-produced intraluminal vesicles in NS5A-ApoE double-positive endosomes.** (A) Huh7-Lunet/ApoE[mT2] cells expressing HCV Core-NS2/E2[eYFP] and containing the subgenomic replicon sgJFH1/NS5A[mCherry] (Fig 5A) were investigated with the CLEM method at 48 h post-electroporation. Lipid droplets stained with lipidTox were used as fiducial markers to correlate light and electron micrographs. Dashed squares in the overlay image (right panel) refer to NS5A-ApoE double-positive structures. The left panels show single-channel light or EM micrographs of the enlarged overlay image on the right. For ease of visualization, endosome peripheries are marked with dashed lines. (B) Magnified views of regions indicated in the dashed squared areas in the overlay image in (A). Cropped areas 1, 2, 3: ApoE-NS5A double-positive ILVs in endosomes; crop 4: an ApoE-positive, NS5A-negative endosome; cropped areas 5 and 6: NS5A-E2 double-positive areas containing numerous DMVs. Orange arrowheads point to ILVs in crops 1, 2 and 3; and DMVs in crops 5 and 6. LD, lipid droplet.

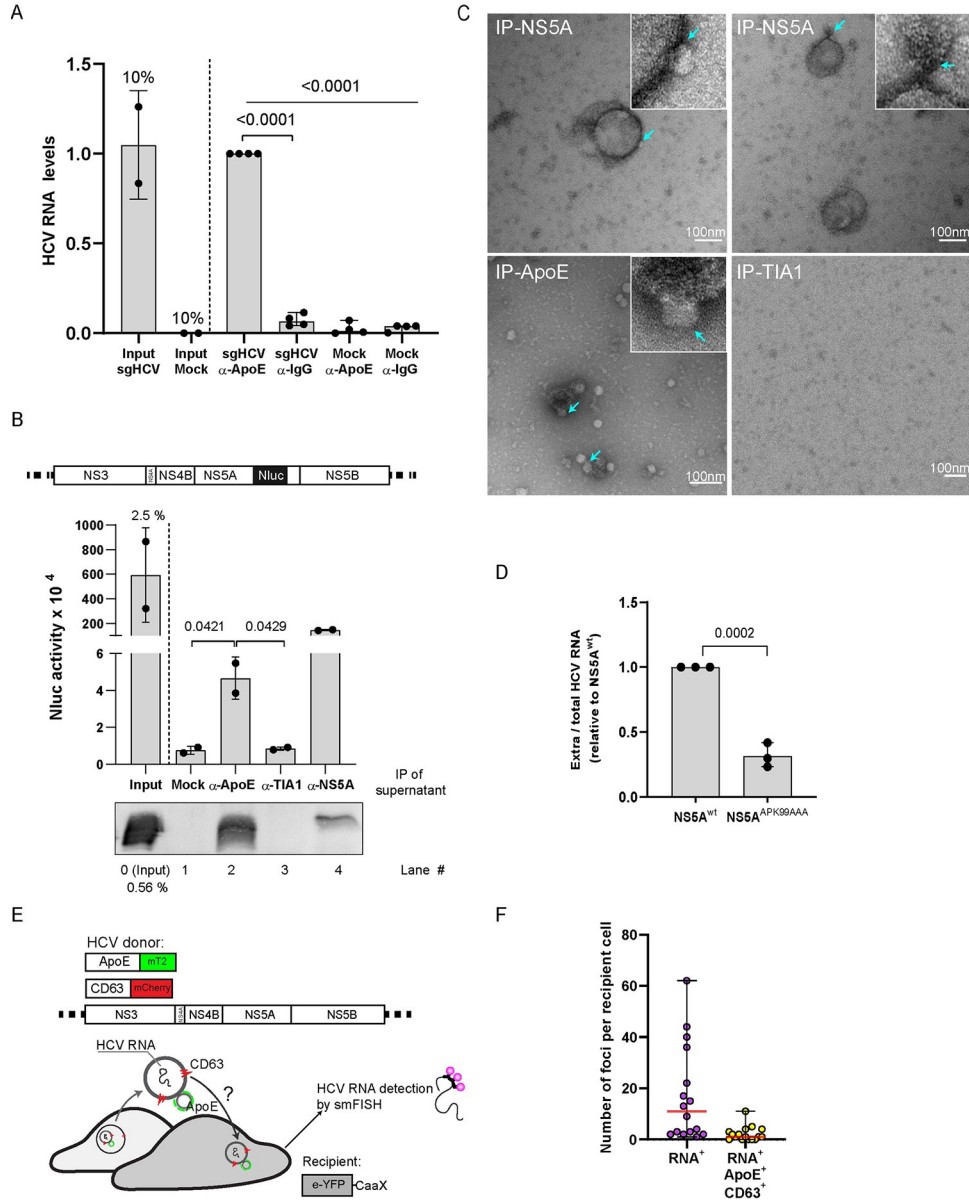

**Fig 8. Association of ApoE-lipoproteins with endosome-derived extracellular vesicles containing HCV NS5A and RNA and evidence for intercellular co-transmission.** (A) Virion-free release of HCV RNA in association with ApoE. Huh7 cells harboring a subgenomic HCV replicon and control cells were cultured in a medium containing 1% FCS for 6 h. Culture supernatants were subjected to immunoprecipitation using ApoE-specific or IgG control antibodies. Immuno-complexes were analyzed by HCV-specific RT-qPCR. Data are means (SD) from 4 independent experiments. P-value was determined using one-way ANOVA and unpaired Student's $t$-tests. (B-C) Association of secreted ApoE with NS5A-containing EVs. (B, top panel) Schematic of the Nanoluciferase (Nluc)-tagged NS5A subgenomic replicon construct. (B, middle and bottom panel) Huh7-Lunet cells were electroporated with subgenomic replicon RNA encoding the NLuc-tagged NS5A and 72 h post-electroporation, culture supernatant was subjected to immunoprecipitation using ApoE-, or NS5A-, or control TIA1-specific antibodies. NS5A contained in captured immuno-complexes was quantified by measuring NLuc activity (middle panel). ApoE contained in captured complexes was analyzed by Western blot (bottom panel). Data are means (SD) from two biologically independent experiments. P-value was determined using unpaired Student's $t$-test. (C) Captured complexes from (B) were visualized by negative staining and analyzed by TEM. Turquoise arrows point to LP-like particles (~20 nm) attached to EVs that were captured with antibodies specified on the top of each panel. (D) Reduced virion-free secretion of HCV RNA with the ApoE-binding deficient NS5A[APK99AAA] mutant. Total RNA contained in Huh7-Lunet cells with stable wildtype or mutant subgenomic replicon was extracted and HCV RNA was quantified by RT-qPCR. In addition, total RNA in culture supernatants was isolated and HCV RNA contained therein was quantified by RT-qPCR. Ratios of secreted to total HCV RNA are shown. Data are means (SD) from three independent experiments. P-value was

determined using unpaired Student's *t*-test. (E-F) Uptake of ApoE-associated, virion-free released HCV RNA by HCV-negative bystander cells. (E) Experimental approach. Huh7-Lunet cells expressing tagged ApoE and CD63 and containing a subgenomic replicon (constructs on the top) served as donor cells. Huh7-Lunet-derived recipient cells expressed eYFP, fused to the farnesylation signal from human HRAS protein (CaaX) to visualize cellular membranes. Donor and recipient cells were co-cultured for 24 h, fixed, and HCV RNA in recipient cells was detected by using smFISH with Hulu probes. (F) The number of total HCV RNA positive structures and ApoE-CD63 double-positive HCV RNA containing structures detected in single recipient cells (n = 16) from (E) was quantified using ColocQuant and ColocJ software suit and data are shown as medians (range). Note that non-specific signals in the cell nuclei were excluded from the analysis.

With the aim to visualize intracellular HCV RNA and its association with ApoE, we employed single molecule Fluorescence In Situ Hybridization (smFISH). Used probes were conjugated to Alexa Fluor 647 and enabled visualization of HCV RNA molecules without signal amplification; yet single molecules could not be resolved beyond the diffraction limit inherent to confocal microscopy (S9A Fig). Although we observed strong nuclear background staining, we noted that the cytoplasmic staining of HCV RNAs was specific, because we detected numerous cytoplasmic foci of viral RNA only in replicon cells, but not in the control cells (S9B Fig). To determine if ApoE associates with HCV RNA-containing EVs that might be transferred to neighboring (bystander) cells, we established Huh7-Lunet/ApoE^mT2 cells containing a stable subgenomic HCV replicon and expressing CD63^mCherry (Fig 8E). These cells were used for co-culture experiments and served as donors. As recipient cells, we used HCV-negative Huh7-Lunet cells expressing the CaaX^eYFP membrane sensor. HCV RNAs were found to partially colocalize with ApoE-CD63 double-positive puncta in donor cells (S9C Fig, area 1). Remarkably, we could detect distinct foci of HCV RNA in single recipient cells, ~13% of them being ApoE-CD63 double-positive (example image in quantification in Fig 8F and area 2 in S9C Fig). Moreover, several of these structures were surrounded by the CaaX^eYFP membrane sensor (S9C Fig, circle, right panel), indicating endosomal uptake into recipient cells rather than mere binding to the plasma membrane. Taken together, these data suggest that HCV might hijack the late endosomal trafficking and egress of ApoE-LPs to transmit NS5A and viral RNA via endosome-derived EVs.

## Discussion

In this study, we developed two tags for ApoE labeling that do not impact its function while allowing the tracking of hepatocyte-made ApoE by live-cell imaging and various other imaging modalities. Obtained results suggest that hepatic ApoE-LPs follow the trafficking pathway of CD63-positive late endosomes. This pathway appears to be hijacked by HCV using the multifunctional protein NS5A that binds to ApoE to release EVs containing viral RNA. Our observations suggest that late endosomes in hepatocytes might be a central site for the storage and secretion of ApoE-LPs. Since biosynthesis and secretion of ApoE-LPs such as VLDL depend largely on the availability of dietary fat, and have to respond rapidly to elevated plasma insulin levels by retaining hepatic lipids [72,73], a lipid reservoir like late endosomes would allow rapid response to fluctuating food and insulin levels. Interestingly, non-hepatic cell lines such as HEK293T and Hela do not express any apolipoprotein but upon ectopic expression, ApoE displayed a similar trafficking pathway like in Huh7-derived cells. It is possible that in the absence of ApoB, ApoE associates to lipoproteins as the sole apolipoprotein in these cells [74], but this remains to be investigated.

Several viruses exploit ApoE for their replication cycles. Two prominent examples are HBV and HCV that both associate with ApoE-containing lipoprotein particles [16,32,75,76]. In the case of HCV, ApoE interacts with NS5A and the envelope glycoproteins and these interactions

are critical for HCV particle assembly and maturation [32,60,61]. Here, we provide evidence that ApoE-NS5A interaction is additionally required for the secretion of HCV-induced EVs containing viral RNA. Release of HCV NS5A and virion-free RNA has been suggested in several independent studies [33–36,62,63,77], but the role of ApoE in this process has not been studied. Our results suggest that ApoE is a critical component for the release of EVs from HCV-replicating cells and these vesicles can be transmitted to naïve bystander cells, consistent with the virion-free transfer of intact HCV genomes from cell to cell [33–36].

Our results address another long-standing conundrum in HCV biology, i.e., the tight association between ApoE and NS5A [59–61]. Although both proteins localize to opposing sites of the ER membrane [27,78], we can efficiently capture EVs from HCV-replicating cells by NS5A pull-down, indicating that NS5A resides on the surface of EVs where it can interact with ApoE. How NS5A might end up on the surface of these vesicles is not known. Previous studies of poliovirus replication have shown that the viral replicase machinery resides on the surface of replication vesicles, which bear resemblance to the DMVs induced by HCV [79–81]. Assuming a poliovirus-like replication mechanism for HCV would explain the localization of NS5A on the outside of the DMVs. Regarding functional relevance, we note that the ApoE-NS5A interaction is not required for HCV virion assembly, at least in the subgenomic replicon model, but appears to boost the release of viral RNA from infected cells, e.g. to avoid recognition by innate RNA sensors such as TLR3 [62].

EVs are phospholipid bilayer-enclosed structures released from cells and containing various signaling molecules [82–85]. They are considered as a "language" exploited by cells and viruses for intercellular communication [86–89]. Several lines of evidence argue for interaction between LPs and EVs. First, various procedures of EV isolation and purification, including size and density fractionation as well as enrichment of CD63-positive EVs do not allow complete separation of LPs and EVs [55–57,90]. Second, LPs were found to attach *in vitro* to purified EVs or even fuse to crude EVs in blood plasma [19,20,91,92]. Third, pigment cell-derived ApoE associates with endosome-derived ILVs and plays an important role in the sorting of a distinct cargo to ILVs and its release via exosomes [29]. Although these studies suggest an association of LPs with ILVs/EVs, to the best of our knowledge, the association between liver-generated LPs and endosome-derived EVs is not well documented and their possible intercellular co-transmission has been unknown.

Our data suggest that in naïve and HCV-infected hepatocytes, ApoE-LPs and endosome-derived CD63-positive ILVs/EVs share a common intracellular late endosomal trafficking in a microtubule dependent and autophagy independent manner. In addition, they appear to be co-secreted and internalized into bystander cells. Although we cannot rule out that these two particle species are separately secreted and associate at later stages in the extracellular milieu or during endocytosis, we postulate that at least a fraction of ApoE-CD63 double-positive structures form intracellularly and are co-secreted. This assumption is based on the observation that microtubule depolymerization concomitantly reduced the deposition of ApoE-CD63 double-positive complexes at the cell periphery and mitigated their secretion, implying that ApoE- and CD63-containing structures within endosomes might be co-secreted. This is in line with our finding using the CD63$^{pHluorin}$ construct showing the secretion of ApoE-containing CD63-positive ILVs. While further work will be needed to provide more direct evidence for this hypothesis, our data argue for a stable interaction between ApoE-LPs and CD63-positive ILVs/EVs. This would explain the difficulty to separate LPs from EVs [55–57,90], which poses a major challenge to assign distinct functions to each of these vesicle species individually [93]. The mechanism underlying this interaction is unknown, but might be mediated by associations between ApoE on LPs and scavenger receptor class B type 1 (SR-BI) or heparan sulfate decorating the surface of ILVs/EVs [92,94]. These interactions could also modulate lipid

transfer from LPs to ILVs/EVs [92]. Moreover, since hepatic ApoE-LPs are secreted into the blood stream, they might alter the systemic spread of EVs into different distant tissues and organs, thus manipulating various biological responses depending on EV content. For instance, the amount of liver-generated plasma ApoE was found to be associated with unfavorable alterations in neurodegenerative diseases including synaptic integrity [95]. The underlying mechanism has not been determined but might be due to the direct contribution of ApoE to lipid metabolism or ApoE-facilitated blood-brain barrier passage of EVs [96–99]. Another example is COVID-19 where plasma-derived EVs isolated from COVID-19 patients alter multiple signaling pathways [100], which might contribute to the broad spectrum of clinical symptoms [101]. Importantly, COVID-19 derived EVs preparations contain multiple apolipoproteins including ApoE, ApoB, ApoA2, ApoD, and ApoH [100].

Our study has some limitations. It is primarily based on the use of human hepatoma cells that are highly permissive to HCV and easy to manipulate. However, because LPs and ILV/EV profiles in vivo are somewhat different, future studies require more physiologically relevant systems, which are however, not permissive to HCV and difficult to manipulate. In addition, although the HCV subgenomic replicon model allows excluding the transmission of HCV RNA via virions, HCV-produced ILVs/EVs might also contain viral structural proteins including the envelope glycoproteins E1 and E2, potentially assisting in the spread of these vesicles [102]. Finally, the physiological consequences of co-spread of hepatic LPs with -EVs in general and in the context of HCV infection, the latter possibly allowing HCV RNA spread independent of virus particles [33–36] remain to be determined but they are beyond the scope of the present study.

In conclusion, our study provides insights into the endosomal egress and transmission of hepatocyte-derived ApoE-containing LPs and the strategy of how HCV exploits this pathway. Given the more general role of EV-mediated cell-to-cell communication, the association of ApoE-LPs with EVs reported here provides new starting points for research into the pathophysiology of ApoE-related metabolic and infection-related disorders.

## Materials and methods

### Materials

Reagents and resources used in this study are provided in S1 Table.

### Methods

**Cell lines and culture conditions.**   All cells used in this study were cultured in Dulbecco's modified Eagle medium (DMEM, Thermo Fisher Scientific), supplemented with 2 mM L-glutamine, nonessential amino acids, 100 U/ml of penicillin, 100 μg/ml of streptomycin, 10% fetal calf serum (DMEMcplt), and given concentrations of antibiotics to select for stable expression of genes of interest. Huh7-Lunet/CD81H cells (750 μg/ml G418) derived from the Huh7 subclone Huh7-Lunet [103] and expressing high levels of the HCV entry receptor CD81, and Huh7-Lunet/CD81H/ApoE-KD cells (5 μg/ml puromycin) with a stable knockdown of ApoE have been described earlier [32,38]. For reasons of simplicity, in this study Huh7-Lunet/CD81H cells are designated Huh7-Lunet cells. HEK293T-miR122 cells (2 μg/ml puromycin), kindly provided by Thomas Pietschmann, have been reported elsewhere [40]. Huh7.5 and HEK293T cells have been described elsewhere [104,105]. HEK293T-miR122, Hela Kyoto, and Huh7-Lunet/ApoE-KD cells were used to generate ApoE$^{mT2}$ expressing cells by lentiviral transduction and stable selection with 10 μg/ml blasticidin. To generate CD63-NLuc expressing cells, Huh7-Lunet cells were transduced with lentiviruses encoding the CD63-Nluc fusion

protein and selected using 500 μg/ml Zeocin. Thereafter, cells were cultured in DMEMcplt containing 50 μg/ml Zeocin.

For the production of HCV-like transcomplemented particles (HCV$_{TCP}$), Huh7-Lunet/ ApoE-KD/ApoE$^{mT2}$ cells (designated Huh7-Lunet/ApoE$^{mT2}$ in this study for reasons of simplicity) were transduced with lentiviruses encoding the HCV structural proteins (C-E1-E2$^{eYFP}$-p7-NS2 or C-E1-E2$^{eYFP}$-p7), selected with 500 μg/ml Zeocin and maintained in 50 ug/ml Zeocin-containing DMEMcplt. To obtain cells with stably replicating subgenomic replicon of the HCV strain JFH1 and used for the coculture experiment, Huh7-Lunet/ ApoE$^{mT2}$/CD63$^{mCherry}$ cells were electroporated with *in vitro* transcripts of the construct sgHyg/JFH1. To monitor HCV RNA secretion in the context of an ApoE-binding defective NS5A mutant or wildtype NS5A, Huh7-Lunet cells were electroporated with *in vitro* transcripts of the construct sgHyg/JFH1/NS5A$^{APK99AAA}$ or sgHyg/JFH1, respectively. Stable cells were selected in a medium containing 400 μg/ml hygromycin and maintained in 150 μg/ml hygromycin-containing DMEMcplt. FCS devoid of extracellular vesicles (EVs) was prepared as previously described [62]. Knock-out (KO)-control and ATG5- and ATG16L1-KO cells were cultured as described elsewhere [70]. The full names of constructs used in this study are given in S1 Table.

**Antibodies and immunofluorescence reagents.** All antibodies and immunofluorescence reagents used in this study are listed in the S1 Table.

**DNA plasmid constructs.** The lentiviral construct pWPI_ApoE encoding human ApoE3 was described previously [106]. To generate pWPI_ApoE$^{FP}$ and pWPI_ApoE$^{SNAPf}$ constructs, the FP- and the SNAPf-coding sequences were amplified by PCR using the corresponding plasmids as templates (see S1 Table) and inserted at the 3' end of the ApoE-coding sequence via the linker sequence SGGRGG. Construct pWPI_CD63$^{mCherry}$ and pWPI_CD63$^{Nluc}$ encode a fusion protein of human CD63 and C-terminal mCherry or NanoLuciferase, respectively. To generate the construct pWPI_eYFP-CaaX, the eYFP-coding sequence was extended at the 3' end by the CaaX coding sequence derived from the human HRAS protein and inserted into the lentiviral vector pWPI. To generate pWPI_CD63_M153R_pHluorin, the CD63-pHluorin coding sequence contained in plasmid pCMV-Sport6-CD63-pHluorin [54] was amplified by PCR and inserted into the lentiviral vector pWPI. To stabilize pHluorin and increase signal intensity, we inserted the M153R mutation [53] by using PCR and primers carrying the desired nucleotide substitutions.

The full-length HCV constructs Jc1 and JcR2A have been described elsewhere [42,107]. The lentiviral constructs encoding the HCV structural proteins Core-NS2/E2$^{eYFP}$ or Core-p7/ E2$^{eYFP}$ were created by replacing the eGFP-coding sequence reported previously [65] by the eYFP-coding sequence. Plasmid pFK_I389neoNS3-3′_dg_JFH1_NS5A-aa2359_mCherry_NS3-K1402Q (designated sgNeo/JFH1/NS5A$^{mcherry}$ in this study) has been reported earlier [108]. To generate the subgenomic replicon encoding a CLIPf-tagged NS5A and the neomycin resistance gene (construct sgNeo/JFH1/NS5A$^{CLIPf}$), the mCherry-coding sequence in construct sgNeo/JFH1/NS5A$^{mcherry}$ was replaced by the CLIPf-coding sequence. To allow selection with hygromycin, the neomycin resistance gene was replaced by the hygromycin resistance gene. To generate the subgenomic replicon construct encoding a NanoLuciferase-tagged NS5A (sgHyg/JFH1/NS5A$^{Nluc}$), the mCherry-coding sequence of construct sgHyg/ JFH1/NS5A$^{mCherry}$ was replaced by the NanoLuciferase-coding sequence [109]. The mutations in NS5A interfering with ApoE interaction (APK99AAA) [60,110] were inserted into the replicon construct sgHyg/JFH1 by using PCR-based mutagenesis.

To generate plasmids encoding myc-tagged NS5A wildtype and the APK99AAA mutant corresponding plasmids were used as template for PCR using primers encoding the myc-tag

sequence and NS5A sequences were inserted into the pCDNA3$^+$ vector. Other plasmids used in this study are listed in the S1 Table.

**Preparation of *in vitro* transcripts and electroporation of HCV RNA.** HCV RNA preparations generated by *in vitro* transcription and transfection of cells by electroporation have been described elsewhere [111]. In brief, plasmids containing HCV JFH1 genomes were linearized using the restriction enzyme MluI-HF (NEB) and purified using the NucleoSpin Extract II Kit (Macherey-Nagel). RNA transcripts were synthesized via *in vitro* transcription using T7 RNA polymerase in 100 μl-reaction mixtures [80 mM HEPES (pH 7.5), 12 mM MgCl$_2$, 2 mM spermidine, 40 mM dithiothreitol, 3.125 mM of each rNTP, 1 U/μl RNasin (Promega), 0.6 U/μl T7 RNA polymerase, and the respective linearized DNA template]. After 4 h at 37˚C, the DNA template was degraded by 45 min treatment with 2 U of RNase-free DNase (Promega) per 1 μg DNA at 37˚C. RNA was purified by acidic phenol-chloroform extraction, precipitated with isopropanol, and dissolved in RNase-free water. The integrity and concentration of RNA were evaluated using agarose gel electrophoresis and spectrophotometry.

For electroporation, confluent cell monolayers were trypsinized and resuspended in Cytomix [120 mM KCl, 0.15 mM CaCl$_2$, 10 mM potassium phosphate buffer, 25 mM HEPES (pH 7.6), 2 mM EGTA, and 5 mM MgCl$_2$] [112] containing 2 mM ATP and 5 mM glutathione (1-2x10$^7$ cells/ml). *In vitro* transcripts (5 μg) were mixed with 200 μl of the cell suspension and electroporation was performed at 975 μF and 166 V using the Gene Pulser system (Bio-Rad) and a cuvette with a gap width of 2 mm (Bio-Rad). Alternatively, 10 μg *in vitro* transcripts were mixed with 400 μl of the cell suspension and electroporation was performed at 975 μF and 270 V using a cuvette with a gap width of 4 mm. After electroporation, cells were immediately transferred to DMEMcplt and seeded into the desired cell culture dishes.

**Western blot analysis.** Cell extracts were prepared using 2x sample buffer [120 mM Tris-HCl (pH 6.8), 60 mM SDS, 100 mM DTT, 1.75% glycerol, 0.1% bromophenol blue] supplemented with 5 mM MgCl$_2$ and 5 U/ml benzonase. Samples were denatured by heating to 95˚C for 5 min. Proteins were separated by SDS-PAGE and transferred to a polyvinylidene difluoride (PVDF) membrane that was blocked by incubation in 5% skim milk-containing PBS-0.05% Tween 20, pH 7.4 (PBST) for 1 h at room temperature (RT). The membrane was incubated with a primary antibody in 1% skim milk-containing PBST for either 1 h at RT or overnight at 4˚C and subsequently incubated with a secondary antibody conjugated with horseradish peroxidase (HRP) for 1 h at RT. Bound secondary antibodies were detected using the Western Lightning Plus-ECL reagent (PerkinElmer) and signals were visualized by using the Intas ChemoCam Imager 3.2 (Intas).

**Quantitative detection of HCV RNA by RT-qPCR.** Total RNA contained in cell lysates or cell culture supernatant was extracted using the NucleoSpin RNA extraction kit (Macherey-Nagel) according to the instruction of the manufacturer. HCV RNA copy numbers in extracted samples were determined with HCV-specific primers and a probe by using the Quanta BioSciences qScript XLT One-Step RT-qPCR KIT (Quanta Biosciences, Gaithersburg, MD) as described elsewhere [62]. Serially diluted HCV *in vitro* transcripts were included in parallel to calculate HCV RNA copy numbers contained in analyzed samples.

**Quantification of ApoE by Elisa.** Extracellular ApoE protein amounts in cell culture supernatants were quantified using a human ApoE ELISA kit (Cell Biolabs, USA) according to the manufacturer's instructions.

**Quantification of HCV Core protein.** Cell culture supernatants collected 24, 48, and 72 h post- electroporation of HCV RNA were filtered through a 0.45 μm pore-size filter (MF-Millipore), inactivated in Triton X-100-containing PBS and diluted at least 10 times in PBS. HCV core protein amount contained therein was quantified using a commercial Chemiluminescent

Microparticle Immunoassay (CMIA) (6L47, ARCHITECT HCV Ag Reagent Kit, Abbott Diagnostics) as reported earlier [65].

**Production of lentiviruses.**   Lentiviruses encoding genes of interest were produced as described recently [113]. In brief, HEK-293T cells were co-transfected with the human immunodeficiency virus-Gag packaging plasmid pCMV-dR8.91, the vesicular stomatitis virus-G encoding plasmid pMD2.G, and the pWPI construct containing the gene of interest by using polyethylenimine (Polysciences Inc.). Lentivirus-containing supernatants were harvested at about 48 h post-transfection and filtered through a 0.45 µm pore-size filter (MF-Millipore).

**Live-cell time-lapse confocal microscopy.**   Cells were seeded onto either 4-compartment (CELLview, Greiner BIO-ONE) or 1-compartment (MatTek Corporation) 35 mm-diameter glass-bottom imaging dishes. Prior to imaging, cells were washed twice and cultured in phenol red-free DMEMcplt. Live-cell time-lapse confocal microscopy was performed in a humidified incubation chamber at 37°C and 5% $CO_2$ using a PerkinElmer UltraVIEW Vox Spinning-Disk microscope equipped with Yokogawa CSU-X1 spinning disk head, Nikon TiE microscope body, a Hamamatsu C9100-23B EM-CCD camera, an automated stage, and the Perfect Focus System (PFS). An Apo TIRF 60x/1.49 N.A. oil immersion objective was used. Multichannel images were acquired sequentially using solid state lasers with excitation at 445 nm for mTurquoise2, 488 nm for pHluorin, 514 nm for eYFP, 561 nm for CLIP^ATTO590, 640 nm for SNAP-^SIR647, and matching emission filters. For imaging of pHluorin-tagged CD63 expressing cells, the medium was supplemented with 25 mM Hepes (pH 7.4) to stabilize neutral pH. The imaging time interval of each experiment is specified in the figure legends.

**Immunofluorescence staining and confocal microscopy.**   Immunofluorescence (IF) staining was performed as previously described [32]. Briefly, cells seeded onto coverslips were fixed with 4% paraformaldehyde (PFA) in PBS for 10 min at RT and permeabilized with 0.1% Triton X-100 in PBS for 10 min at RT. After blocking with 3% (w/v) bovine serum albumin (BSA) in PBS for 20 min at RT, cells were incubated with a diluted primary antibody in 1% BSA/PBS for 1 h at RT or overnight at 4°C. Cells were further incubated with a diluted secondary antibody conjugated with Alexa Fluor 488 or 568, or 647 (1:1000) in 1% BSA/PBS (Molecular Probes) for 1 h in the dark at RT (see S1 Table). If required, cell nuclei were counterstained with DAPI (1:3000) (Molecular Probes). In-between each step, cells were washed at least 3 times with 1x PBS. Unless otherwise stated, coverslips were mounted with Fluoromount-G mounting medium (Electron Microscopy Sciences, Ft. Washington, USA) overnight at 4°C. For selective permeabilization assay, cells were permeabilized in 5 µg/ml digitonin dissolved in PBS for 15 min at 4°C. IF images were generated with a spinning-disk confocal microscope (PerkinElmer). The fluorescence intensity profiles among channels were obtained with Fiji using the "Plot Profile" function. While raw values of one selected channel were kept unchanged, the values of the other channels along the same line were normalized by multiplying them with the quotient of the mean intensity of the selected channel to that of the remaining channels.

**Super-resolution microscopy.**   Huh7-Lunet/ApoE^SNAPf cells stably expressing ApoE^SNAPf were electroporated with *in vitro* transcripts of HCV sgNeo/JFH1/NS5A^CLIPf and grown on high precision glass coverslips (Deckglaeser, Marienfeld). At 48 h post-electroporation, cells were sequentially incubated with CLIP^ATTO590 (1:2500) and 5 µM SNAP^SiR647 (NEB) in DMEMcplt for 1 h. Cells were washed intensively at least 3 times with DMEMcplt and cultured for 15 min. Thereafter, cells were washed 3 times with PBS, fixed with 4% PFA in PBS for 10 min at RT, and subjected to immunofluorescence staining using anti-CD63 antibody conjugated to Alexa Fluor 488 (Santa Cruz). Cells were mounted with ProLong Gold Antifade Mountant (ThermoFisher Scientific) by overnight incubation at RT. STED imaging was conducted using an Expert Line STED system (Abberior Instruments GmbH, Göttingen,

Germany) equipped with an SLM based easy3D module, an Olympus IX83 microscope body, solid state pulsed lasers (488 nm, 590 nm, and 640 nm), and the 775 nm STED laser. The 100x oil immersion objective (NA, 1.4; Olympus UPlanSApo) was used. Initially, confocal images were captured in the line sequential mode using the following excitation lasers: 488 nm for AF488, 590 nm for ATTO590, 640 nm for SIR647, and the corresponding 525/50, 615/20, and 685/70 emission filters. These filters are placed in front of avalanche photodiodes for detection. Small regions of interest were selected and subjected to STED imaging. STED images in selected areas were captured sequentially using the 590 nm and 640 nm excitation laser lines in the line sequential mode with corresponding 615/20 and 685/70 emission filters, followed by the depletion using the 775 nm STED laser. STED images were deconvoluted using the Huygens Deconvolution software (Scientific Volume Imaging) using Classic Maximum Likelihood Estimation (CMLE) algorithm and Deconvolution Express mode with "Conservative" settings.

**HCV RNA detection by single-molecule fluorescence in situ hybridization.** Intracellular HCV RNAs were visualized by smFISH using Hulu probes (PixelBiotech, Germany) according to the manufacturer's instruction with slight modifications. In brief, cells grown on glass coverslips were fixed with 4% paraformaldehyde (PFA) in PBS for 30 min at RT. Cells were then treated with 150 mM glycine in PBS to quench residual PFA, permeabilized by treatment with 0.1% Triton X-100 in PBS for 10 min, and incubated with proteinase K (1:4000) (ViewRNA ISH Kit, ThermoFisher Scientific) in PBS for 5 min. Concentration of proteinase K and treatment duration were chosen to improve smFISH probe signal/noise ratio while minimizing signal loss of the fluorescent protein. HCV RNAs were hybridized to Hulu probes targeting the positive strand in the NS3 coding region (nucleotides 3733–4889 of the JFH-1 genome; GenBank accession number AB047639). Hybridization was done in HuluHyb solution (2xSSC, 2 M Urea, 10% dextran sulfate, 5x Denhardt's solution) using a humidified chamber at 30°C overnight. Cells were washed extensively with HuluWash and coverslips were mounted on glass slides with Prolong Gold Antifade Mountant (ThermoFisher Scientific) by overnight incubation at RT.

**Immunoprecipitation.** HEK293T-miR122 cells were co-transfected with HA-tagged ApoE construct, or an empty vector, or pCDNA3+ myc-tagged NS5A[wt], or myc-tagged NS5A[APK99AAA], respectively, using the TransIT-LT1 Transfection Reagent (Mirus Bio). After 30 h, cells were lysed by 10 min incubation in lysis buffer [50 mM Tris-HCl, pH 7.5, 150 mM NaCl, 1% Triton X-100, 1 mM EDTA, 10% glycerol, 1x protease inhibitor cocktail (Roche)] on ice. Cell lysates were centrifuged at 15,000 x *g* for 15 min at 4°C. Cleared supernatants were incubated with protein G-magnetic bead slurry (Dynabeads, ThermoFisher Scientific) for 30 min at 4°C to remove proteins binding to the resin. Beads were removed by pelleting with a magnetic stand and supernatants were incubated with rabbit anti c-myc antibody at 4°C overnight. Protein complexes were captured by adding protein G bead slurry and 1 h incubation of samples under continuous rotation at 4°C. Beads were washed 5 times with lysis buffer lacking glycerol, captured protein complexes were eluted with 2x sample buffer and denatured for 5 min at 95°C. Proteins were analyzed by Western blot using mouse anti-HA antibody.

**Iodixanol density gradient centrifugation.** Cells were washed and cultured for 5 h in 1% FCS-containing DMEM. Thereafter, cell culture supernatant was filtered through a 0.45 μm pore-size filter (MF-Millipore), loaded on top of a PBS-based 10–50% iodixanol gradient (Sigma Aldrich), and subjected to isopycnic centrifugation for 18 h at 34,000 rpm (~120,000 x *g*) at 4°C using an SW60 rotor (Beckman Coulter, Inc.). Eleven fractions were collected from top to bottom and analyzed by density measurement using a refractometer (Kruess, AGS Scientific) and Western blot.

**Luciferase reporter assay.** HCV RNA replication kinetics were determined by using the HCV JcR2A reporter construct. Briefly, cells were collected at 4, 24, 48 and 72 h post-

electroporation and lysed in luciferase lysis buffer (1% Triton X-100, 10% glycerol, 25 mM gly-cylglycine, 15 mM $MgSO_4$, 4 mM EGTA, and 1 mM DTT) for 15 min at RT. Cell lysates were transferred to 96-well plates and coelenterazine-containing luciferase assay buffer (25 mM gly-cylglycine, 15 mM $MgSO_4$, 4 mM EGTA, 1 mM DTT, and 15 mM K3PO4, pH 7.8) was injected. Renilla luciferase activities were measured using a Mithras LB 940 plate luminometer (Berthold Technologies, Freiburg, Germany). Obtained values were normalized to the 4 h value of each transfection to correct for transfection efficiency. To measure the transmission of HCV, culture supernatants were used to inoculate naïve Huh7.5 cells, and after 72 h, cells were lysed and subjected to luciferase assay. NanoLuciferase (Nluc) activity was measured using the Nano-Glo Luciferase Assay System (Promega) according to the instruction of the manufac-turer with slight modifications. In brief, 50 µl of samples were mixed with 50 µl NLuc substrate (1:1000) in the assay buffer and NLuc activities were measured using a Mithras LB 940 plate luminometer (Berthold Technologies, Freiburg, Germany).

**Immunocapture of extracellular ApoE-associated structures.** Supernatants of cells cul-tured in EV-free DMEM were collected, filtered through a 0.45 µm pore-size filter (MF-Milli-pore), and incubated with an anti-ApoE antibody for 3 h at 4˚C. ApoE-associated structures were captured using protein G-magnetic beads (Dynabeads, ThermoFisher Scientific) and overnight incubation at 4˚C with continuous rotation. After 5 times washing with ice-cold PBS, protein complexes were eluted by 10 min incubation with 0.1 M glycine, pH 2.5 at RT, and samples were neutralized by adding 1 M Tris, pH 7.5.

**Transmission electron microscopy and correlative light and electron microscopy (CLEM).** Sample preparation, data acquisition, and data processing were conducted as described earlier [65] with slight modifications. For CLEM, cells were fixed for 30 min at RT with a fixative containing 0.2% glutaraldehyde (GA) and 4% PFA and then washed 3 times with PBS to remove the fixative. The coordinates of cells-of-interest on the gridded MatTek dish were captured with the 20x objective using transmitted light with differential interference contrast (DIC). Cells were then subjected to immunofluorescence imaging using an oil immer-sion 60x objective, covering the ~2.8 µm cell thickness with 0.2 µm spacing between optical planes before and after the addition of LipidTox Deep Red Neutral Lipid Stain (Invitrogen). Cells were further postfixed in 2.5% GA in CaCo buffer with supplemented ions [2.5% GA, 2% sucrose, 50 mM sodium cacodylate (CaCo), 50 mM KCl, 2.6 mM $MgCl_2$, and 2.6 mM $CaCl_2$] for 30 min or overnight at 4˚C. After 3 washes with 50 mM CaCo buffer, cells were incubated with 2% osmium tetroxide in 50 mM CaCo for 40 min on ice, washed 3 times with milli-Q water, and incubated with 0.5% uranyl acetate in water at 4˚C. Samples were washed again with water prior to the sequential dehydration of cells using a graded ethanol series from 50% to 100% at RT. Samples were embedded in Epon 812 (Carl Roth) and incubated for at least 2 days at 60˚C to allow polymerization of the resin. Epon was detached from the glass coverslips by dipping it several times into liquid nitrogen followed by hot water. Cells of interest were identified by the negative imprint of the gridded coverslips and cut into 70 nm ultrathin sec-tions using an ultramicrotome (Leica EM UC6, Leica Microsystems). Sections were collected on pioloform coated copper palladium slot grids (Science Services, GMBH) and counter-stained sequentially with 3% uranyl acetate in water for 5 min and lead citrate (Reynold's) for 5 min. Images were acquired by using the Jeol JEM-1400 (Jeol Ltd., Tokyo, Japan) transmis-sion electron microscope (TEM) equipped with a 4k pixel digital camera (TemCam F416; TVIPS, Gauting, Germany) and the EM-Menu or Serial EM software [114]. Lipid droplets were used as fiducial markers to correlate the EM with the light micrographs using the Land-mark Correspondences plugin in the Fiji software package [115]. To visualize ApoE-contain-ing structures enriched by immunocapture, samples were added onto freshly glow-discharged

carbon- and pioloform-coated 300-mesh copper grids (Science Services GmbH, Munich, Germany) and subjected to negative staining using 3% uranyl acetate for 5 min at RT.

**Immunogold labeling.** For immunogold labeling of ApoE-associated structures, all incubation and washing steps were conducted by floating the grids on top of drops at RT. In-between each step, samples were washed at least 5 times for 2 min with PBS. The basic protocol employed has been reported elsewhere [116] and only slight modifications were made. In brief, samples absorbed onto copper grids were blocked with the blocking solution [0.8% BSA (Roth, Karlsruhe, Germany), 0.1% fish skin gelatin (Sigma-Aldrich), 50 mM glycine in PBS]. For ApoE and CD63 labeling, grids were incubated with goat anti-ApoE antibody (1:100) and mouse anti-CD63 antibody (1:100) in blocking solution, respectively, for 30 min at RT. Grids were further incubated with rabbit anti-goat- or anti-mouse-bridging antibody (1:150) in the blocking solution for 20 min. Bound antibodies were detected with protein A conjugated to 10-nm gold particles diluted 1:50 in blocking buffer for 30 min. Grids were fixed with 1% glutaraldehyde in PBS for 5 min, washed 7 times with $H_2O$, briefly rinsed with 3% uranyl acetate, and negatively stained again with 3% uranyl acetate for at least 5 min.

**Automated particle tracking in fluorescence microscopy images.** Particle tracking in fluorescence microscopy images was performed by using a probabilistic particle tracking approach that is based on Bayesian filtering and multi-sensor data fusion [117]. This approach combines Kalman filtering and particle filtering and integrates multiple measurements by separate sensor models as well as sequential multi-sensor data fusion. The sensor models determine detection-based and prediction-based measurements via elliptical sampling [118] and take into account different uncertainties. In addition, the tracking approach exploits motion information by integrating displacements in the cost function for correspondence finding. Particles are detected by the spot-enhancing filter (SEF) [119] consisting of a Laplacian-of-Gaussian (LoG) filter followed by intensity thresholding of the filtered image and determination of local maxima.

**Quantification of object-based colocalizations in multi-channel fluorescence microscopy images.** Donor and recipient cell areas were manually segmented based on cell peripheries visualized by plasma membrane-resident fluorescent signals of CD63$^{mCherry}$ and membrane marker CaaX$^{eYFP}$ using the Fiji software package. Object-based single protein detection and colocalizations in two- or three-channel microscopy images in regions of interest were quantified using the ColocQuant and ColocJ software suite, which employ a graph-based k-d-tree approach that efficiently computes nearest neighbor queries based on Euclidean distances [120].

**Motility analysis of ApoE$^{mT2}$ and CD63$^{mCherry}$.** The motility of ApoE$^{mT2}$- and CD63$^{mCherry}$-positive puncta was quantified by a mean squared displacement (MSD) analysis [121] using the computed trajectories. For each trajectory with a minimum of 10 time points (corresponding to a time duration of 32.5 s), we computed the MSD as a function of the time interval $\Delta t$. All MSD curves corresponding to ApoE and CD63 respectively were averaged to obtain the respective MSD curves. To quantify the motility, we fitted the anomalous diffusion model $MSD(\Delta t) = 4\Gamma \Delta t^{\alpha}$ to the MSD values and obtained the anomalous diffusion exponent $\alpha$ for motion classification and the transport coefficient $\Gamma [\mu m^2 s^{-\alpha}]$. The motion of ApoE$^{mT2}$ and CD63$^{mCherry}$ was classified into confined diffusion ($\alpha \leq 0.1$), obstructed diffusion ($0.1 < \alpha < 0.9$), normal diffusion ($0.9 \leq \alpha < 1.1$), and directed motion ($\alpha \geq 1.1$) [122]. To quantify the diffusion coefficient $D [\mu m^2 s^{-1}]$, we fitted the normal diffusion model $MSD(\Delta t) = 4D\Delta t$ to the MSD values.

Automatic colocalization of ApoE$^{mT2}$ and CD63$^{mCherry}$ was performed using the computed trajectories with a minimum of 10 time points (corresponding to a time duration of 32.5 s). For each time point, colocalization was determined using a graph-based k-d-tree approach,

which efficiently computes a nearest neighbor query based on Euclidean distances. An ApoE particle is considered to be colocalized with a CD63 particle, if the ApoE particle has a nearest CD63 particle within a maximum distance for at least a minimum number of consecutive frames. Otherwise, the ApoE particle is considered as non-colocalized with a CD63 particle. We used a maximum distance of 5 pixels (corresponding to 0.449 μm) and a minimum number of four consecutive frames (corresponding to 13 s). The computed colocalization information was visualized by color representations, and the motility of colocalized and non-colocalized ApoE was quantified by an MSD analysis.

To quantify the directed motion of colocalized ApoE$^{mT2}$ and CD63$^{mCherry}$, we performed a MSD analysis [121] using the computed colocalized trajectories of these proteins. To robustly classify the motion type into directed and non-directed motion of colocalized ApoE, we fitted for each trajectory the anomalous diffusion model $MSD(\Delta t) = 4\Gamma\Delta t^{\alpha}$ to the MSD values in two intervals from $\Delta t = 0$ s to 25 s and from $\Delta t = 0$ s to 60 s. Directed motion is considered if for one of the intervals we have $\alpha \geq 1.1$, otherwise non-directed motion is considered. For the classified trajectories, the MSD curves were averaged to obtain an MSD curve for colocalized ApoE with directed and non-directed motion, and the motion was quantified by the transport coefficient $\Gamma[\mu m^2 s^{-\alpha}]$, the diffusion coefficient $D[\mu m^2 s^{-1}]$, and the anomalous diffusion exponent $\alpha$.

**Quantification and statistical analysis.** Unless otherwise stated, differences between sample populations were evaluated using a two-tailed, unpaired Student's $t$-test or Mann-Whitney test for the image-based analysis provided in the GraphPad Prism 8 software package. Differences with P-values less than 0.05 are considered to be significant and shown on the graph. The sample size of each experiment is specified in the corresponding figure legend.

## Supporting information

**S1 Fig. Functionality of ApoE$^{mT2}$.** (A) Validation of ApoE tagging with various fluorophores and confirmation of expression. Huh7-Lunet cells with stable knockdown (KD) of ApoE were transduced with lentiviruses encoding different fluorescently tagged-ApoE variants. After selection for stable expression, lysates of given cell pools were analyzed by Western blot using an ApoE-specific antibody. α-tubulin served as a loading control. mScarlet-C1: wildtype mScarlet; mScarlet-H: photo-stable mScarlet (M164H) variant. (B) Subcellular distribution of ApoE$^{mT2}$ in HEK293T (left) and Hela cells (right) stably expressing this protein after lentiviral transduction and selection. Cells were characterized by confocal microscopy. (C) Expression of ApoE$^{wt}$ (wt) and ApoE$^{mT2}$ (mT2) in reconstituted Huh7-Lunet cells with stable depletion of endogenous ApoE (Huh7-Lunet/ApoE-KD cells) was examined by Western blot using an ApoE-specific antibody. β-actin served as a loading control. Level of ApoE protein expression relative to wt (set to 1) is given below the lanes. (D-E) HCV replication in Huh7-Lunet/ApoE$^{mT2}$ cells. (D) Cells were transduced with either an empty vector (Empty V), or ApoE$^{wt}$, or ApoE$^{mT2}$, respectively, and selected for stable transgene expression. Cells were then electroporated with *in vitro* transcripts of the HCV Renilla luciferase (RLU)-reporter virus (JcR2a). HCV replication was determined at indicated time points by measuring RLU activities in cell lysates. (E) Amounts of core protein contained in cells from (D) at indicated time points were measured by chemiluminescence assay. Data are means of internal replicates from a representative experiment (n = 3).
(TIF)

**S2 Fig. Functionality of ApoE$^{mT2}$ as determined by rescue of infectious HCV particle production.** (A-B) HCV replication in HEK293T-miR122-ApoE$^{mT2}$ cells. (A) Cells were transduced with either an empty vector (Empty V.), or wildtype ApoE (ApoE$^{wt}$), or ApoE$^{mT2}$,

respectively, and electroporated with *in vitro* transcripts of the HCV Renilla luciferase (RLU)-reporter virus (JcR2a). HCV replication was determined at indicated time points by measuring RLU activities in cell lysates. RLU activities were normalized to the 4 h value to correct for the transfection efficiency. (B) Amounts of core protein contained in cells from (A) at indicated time points were measured by chemiluminescence assay. Data in both panels are means for a representative experiment (n = 2). (C-D) Production of infectious HCV in HEK293T-miR122-ApoE$^{mT2}$ cells. (C) At 24 and 48 h post-electroporation, amounts of extracellular core protein present in supernatants of cells from (A) were determined by chemiluminescence assay. (D) Culture supernatants harvested at 24 and 48 h post-electroporation were used to inoculate naïve Huh7.5 cells and HCV replication therein was measured by quantifying RLU activity at 72 h after inoculation. Virus titers normalized to HCV RNA replication in trans-fected cells are shown. Data in both panels are means for a representative experiment (n = 2).
(TIF)

**S3 Fig. Colocalization of ApoE$^{mT2}$ with Rab7 and ADRP. H**uh7-Lunet/ApoE$^{mT2}$ cells were transduced with lentiviruses encoding Rab7$^{mCherry}$ (upper panel) or ADRP$^{mCherry}$ (lower panel). Cells were fixed and analyzed by confocal microscopy. Boxed areas in the left panels are shown as enlarged views in the panels on the right of each row. Arrowheads point to ApoE-Rab7 positive signals.
(TIF)

**S4 Fig. Colocalization of ApoE$^{mT2}$ with secretory and endocytic markers in non-hepatic cell lines.** HEK293T/ApoE$^{mT2}$ (A) and Hela/ApoE$^{mT2}$ cells (B) were subjected to immunos-taining of markers of the ER (PDI), Golgi (GM130), or transduced with lentiviruses encoding CD63$^{mCherry}$ or Rab7$^{mCherry}$ to label intraluminal vesicles/endosomes, and analyzed by confo-cal microscopy. Boxed areas in the left panels are shown as enlarged views in the panels on the right of each row. Profiles on the right of each panel were taken along the lines indicated with white arrows in cropped images.
(TIF)

**S5 Fig. Newly synthesized ApoE abundantly colocalizes with CD63-positive endosomes and are distinct from secreted and re-internalized ApoE-containing structures.** (A) Huh7--Lunet/ApoE-KD cells were lentivirally transduced with the ApoE$^{mT2}$ expression vector for 4 h (donor). Cells were then washed twice with PBS and Huh7-Lunet/ApoE-KD recipient cells expressing the eYFP$^{CaaX}$ membrane sensor were added. After 12 h, cells were fixed, subjected to immunostaining to label CD63, and analyzed by confocal microscopy. Boxed area in the top panel is shown as an enlarged view in the panels on the bottom. Arrowheads point to newly synthesized ApoE-CD63 double-positive signals in the donor cell. D: donor, R: recipient. (B) Expression and secretion of mTurquoise2-, SNAPf- and KDEL-tagged ApoE. Lysates and cul-ture supernatants of Huh7-Lunet/ApoE-KD cells expressing ApoE$^{mT2}$, or ApoE$^{SNAPf}$, or ApoE$^{mT2-KDEL}$, or ApoE$^{KDEL}$ were analyzed by Western blot using ApoE-specific antibody. β-actin served as loading control. KDEL-tagged ApoE that is retained in the ER served as speci-ficity control to determine ApoE$^{SNAPf}$ secretion. The ratios of secreted to total ApoE for each given construct are displayed below the corresponding lanes. The ratio of ApoE$^{mT2}$ was set to 1. (C) Huh7-Lunet/ApoE-KD cells were lentivirally transduced with the ApoE$^{SNAPf}$ expression vector. After 4 h, cells were cultured in medium containing 5 μM cell-non-permeable SNAP-surface substrate for 12 h to selectively label ApoE$^{SNAPf}$ present in the culture supernatant. Thereafter, cells were fixed, subjected to immunostaining to label total intracellular ApoE and CD63, and analyzed by confocal microscopy. The re-internalized ApoE signals (marked with dashed circle) are visualized via the SNAP-surface substrate and shown in a maximum

intensity Z-projection (1.2 μm, 0.2 μm/step). Boxed area in the top panel is shown as an enlarged view in the panels on the bottom. Arrowheads point to newly synthesized (i.e. SNAP-substrate negative) ApoE-CD63 double-positive signals of a shown focal plane (dashed circles, upper left and right).
(TIF)

**S6 Fig. Secretion of ApoE via the endosomal route.** (A) Huh7-Lunet/ApoE$^{mT2}$ cells expressing CD63$^{mcherry}$ were analyzed by live-cell confocal microscopy. Histograms of sizes of ApoE-CD63 double-positive structures (left) and their trafficking velocities are shown (right). (B) [Left] Mean squared displacement (MSD) of overall ApoE and CD63 trafficking of double-positive structures from (A) and comparison with MSD of those structures displaying directed or non-directed motions. [Right] Example of ApoE-CD63 co-trafficking by directed motion to the cell periphery. Huh7-Lunet/ApoE$^{mT2}$ cells expressing CD63$^{mcherry}$ were analyzed by live-cell confocal microscopy. A maximum projection image showing co-trafficking of an ApoE-CD63 complex with a directed motion to the cell periphery is shown. Frame interval = 2.65 sec; whole duration = 53 sec. (C) Huh7-Lunet cells were either mock-treated or treated with increasing concentration of colchicine for 1 h to depolymerize microtubules. ApoE in cell lysates and in the supernatants was analyzed by Western blot using ApoE-specific antibody; CD63 in cell lysates was analyzed by Western blot using CD63-specific antibody. β-actin served as loading control for cell lysates.
(TIF)

**S7 Fig. Characterization of ApoE variants and NS5A mutants.** (A) ApoE-NS5A colocalization in cells replicating a full-length HCV genome. Huh7-Lunet/ApoE$^{mT2}$ cells were electroporated with *in vitro* transcripts of the HCV genome Jc1. At 54 h post-electroporation, cells were fixed, permeabilized, and incubated with NS5A- and PDI-specific antibodies for subsequent immunofluorescence staining. Images were acquired with a confocal microscope. Arrowheads: ApoE-NS5A signals. Note the high similarity to the structures detected in cells containing the split HCV genome (Fig 5). (B) Example of automated detection and visualization of ApoE-NS5A double-positive puncta from (A) using ColocQuant and ColocJ. Circles and numbers mark the identity of each detected ApoE-NS5A double-positive structure. (C-D) CLIPf-tagged NS5A supports HCV RNA replication. (C) Detection of double-stranded RNA (dsRNA) in cells transfected with sgJFH1/NS5A$^{CLIPf}$. Huh7-Lunet cells electroporated with RNA of sgJFH1/NS5A$^{wt}$, or sgJFH1/NS5A$^{CLIPf}$ or the replication-defective mutant sgJFH1/NS5A$^{wt}$/NS5B$^{delGDD}$ were fixed at 24, 48, and 72 h post-electroporation (p.e.), subjected to immunofluorescent staining of dsRNA, NS5A, and nuclear DNA, and analyzed by wide-field microscopy. Quantitative analysis of dsRNA signal intensity in single cells at indicated time points after normalization to the delGDD control is shown in the right panel. (D) Expression of CLIPf-tagged NS5A. Huh7-Lunet cells were electroporated with RNA of the subgenomic replicon sgJFH1/NS5A$^{wt}$ or sgJFH1/NS5A$^{CLIPf}$, and cell lysates harvested at 24, 48, and 72 h post-electroporation were analyzed by Western blot using NS5A-specific antibody. β-actin served as a loading control. (E) Colocalization of ApoE-NS5A double-positive structure with CD63. Huh7-Lunet/ApoE$^{SNAPf}$ cells were electroporated with subgenomic replicon RNA encoding NS5A$^{CLIPf}$ and after 72 h, cells were sequentially labeled with SNAP$^{SiR647}$ and CLIP$^{ATTO590}$ for 1 h, fixed, permeabilized, incubated with anti-CD63$^{AF488}$ antibody, and subjected to confocal microscopy. Four images on the bottom show single- or merged channels-magnified views of the boxed area in the top overview image. Arrowheads point to ApoE-NS5A-CD63 triple-positive signals. (F) Secretion of NanoLuciferase (Nluc)-tagged NS5A. Huh7-Lunet cells were electroporated with RNA of the subgenomic replicon sgJFH1/NS5A$^{N-luc}$, and cell lysates and supernatants harvested at 24, 48, and 72 h post-electroporation were

subjected to Nluc activity measurement. (G) Mitigation of ApoE-NS5A interaction by a mutation in NS5A domain I. HEK293T-miR122 cells were co-transfected with constructs encoding HA-tagged ApoE and either an empty vector, or myc-tagged NS5A$^{wt}$, or myc-tagged NS5A$^{AP-K99AAA}$, respectively. At 30 h post-transfection, cell lysates were subjected to immunoprecipitation (IP) using a myc-specific antibody and captured complexes were analyzed by Western blot with an HA-specific antibody. Band intensities of co-captured ApoE were quantified and values were normalized to the one obtained with NS5A$^{wt}$ that was set to 1. Total cell lysate (0.5%) was loaded as input.
(TIF)

**S8 Fig. Autophagy-independent enrichment of NS5A in ApoE-containing structures.** (A-B) Unaltered enrichment of NS5A in autophagy-deficient cells. (A) Huh7-Lunet cells with knock-out (KO) of ATG5 or ATG16L1 and control-KO cells were electroporated with HCV subgenomic replicon RNA encoding NS5A$^{mCherry}$. After 54 h, cells were fixed and subjected to immunofluorescence staining of ApoE and LC3 and analyzed by confocal microscopy. Boxed areas in the left panels are shown as enlarged views in the panels on the right of each row. Arrowheads point to ApoE-NS5A-LC3 triple-positive signals. (B) The numbers of ApoE-NS5A double-positive and ApoE-NS5A-LC3 triple-positive signals in single cells from (A) are shown. P-values were determined using Mann-Whitney test. (C) Subcellular distribution of NS5A expressed on its own in relation to ApoE. Huh7-Lunet/ApoE$^{mT2}$ cells were transfected with NS5A expression construct. Cells were fixed at 72 h post-transfection, subjected to immunofluorescence staining of NS5A, and analyzed by confocal microscopy. Boxed area in the top panel is shown as an enlarged view in the panels on the bottom. Plot profile in the right panel is along the line indicated with the white arrow in the lower left crop image.
(TIF)

**S9 Fig. Detection of HCV RNA by single molecule (sm) FISH.** (A) Schematic of the design of smFISH Hulu probes used to detect HCV RNA. These probes target a region encoding for NS3 (nucleotide 3733–4889 of the HCV JFH1 genome; GenBank accession number AB047639). (B) Specificity of HCV RNA detection by smFISH with Hulu probes. HCV RNA contained in Huh7-Lunet cells harboring a subgenomic replicon was detected by smFISH. Huh7-Lunet cells expressing the membrane sensor eYFP-CaaX (farnesylation signal from human HRAS protein) and used as recipient cells in co-culture experiments served as a negative control. The boundaries of cell nuclei were marked with white dashed circles, and the areas within these circles were excluded from the analysis to omit unspecific staining by the RNA probes. (C) Detection of ApoE-associated HCV RNA in recipient cells. Huh7-Lunet/ApoE$^{mT2}$/CD63$^{mCherry}$ cells containing a subgenomic HCV replicon (donor cells) were co-cultured with Huh7-Lunet$^{eYFP-CaaX}$ recipient cells for 24 h. Thereafter, cells were fixed and processed for visualization of HCV RNA by using smFISH. An overview image is shown on the left. Dashed area 1: donor cell; dashed area 2: recipient cell. Magnified views of dashed areas are shown on the right panels. Arrows point to ApoE-associated HCV RNA dots detected in both donor and recipient cells. Arrowheads indicate ApoE-CD63 double-negative HCV RNA dots in the recipient cell.
(TIF)

**S1 Table. Reagent or resource used in this study.**
(DOCX)

**S1 Movie. Intracellular co-trafficking of ApoE-CD63 complexes in an uninfected hepatocyte.** Huh7-Lunet/ApoE$^{mT2}$ cells expressing CD63$^{mcherry}$ were analyzed by live-cell time-lapse confocal microscopy. The trajectories of several ApoE-CD63 double-positive signals are

marked. Frame interval = 3.61 sec. Duration of shown imaging = 111.91 sec.
(AVI)

**S2 Movie. Intracellular co-trafficking of ApoE-CD63 complexes in mock versus colchicine-treated hepatocytes.** Huh7-Lunet/ApoE$^{mT2}$ cells expressing CD63$^{mcherry}$ were mock-treated or treated with 80 μM colchicine for 1 h and analyzed by live-cell time-lapse confocal microscopy. The motility of ApoE and CD63 is shown on the left and right for mock- and colchicine-treated cells, respectively. Frame interval = 3.28 sec. Duration of observation = 118.08 sec.
(AVI)

**S3 Movie. Secretion of an ApoE-associated CD63-positive intraluminal vesicle in an uninfected hepatocyte.** Huh7-Lunet cells expressing ApoE$^{mT2}$ and CD63$^{pHluorin}$ were cultured in imaging medium (pH 7.4) and analyzed by live-cell time-lapse confocal microscopy with a focus on plasma membrane resident CD63-fluorescent signals. Frame interval = 3.14 sec. Duration of shown imaging = 200.96 sec.
(AVI)

**S4 Movie. Uptake of donor-derived ApoE-CD63 complexes by a recipient cell.** Donor (Huh7-Lunet/ApoE$^{mT2}$/CD63$^{mCherry}$) and recipient cells (Huh7-Lunet cells expressing eYFP-tagged CaaX) were co-cultured for 16 h and analyzed by live-cell time-lapse confocal microscopy. An area of a recipient cell (gray) showing the donor-derived ApoE-CD63 double-positive signals is shown. Frame interval = 3.81 sec. Duration of shown imaging: 118.11 sec.
(AVI)

**S5 Movie. Long-term time-lapse confocal imaging of ApoE, NS5A, and E2 trafficking in an HCV-replicating hepatocyte.** Huh7-Lunet cells stably expressing ApoE$^{mT2}$ and C-NS2/E2$^{eYFP}$ were electroporated with the replicon RNA encoding mCherry-tagged NS5A. Cells were subjected to time-lapse live-cell confocal microscopy to monitor ApoE, NS5A, and E2 signals from 5 to 54 h post-electroporation. A duration from 25.5 to 54 h is shown. Frame interval = 30 min.
(AVI)

**S6 Movie. Abundance of ApoE-NS5A foci in an HCV-replicating hepatocyte.** Huh7-Lunet cells stably expressing ApoE$^{mT2}$ and C-NS2/E2$^{eYFP}$ were electroporated with the replicon RNA encoding mCherry-tagged NS5A. At 48 h post-electroporation, cells were subjected to time-lapse live-cell confocal microscopy to monitor ApoE, NS5A, and E2 signals. Frame interval = 10.0 sec. Duration of shown imaging: 490.0 sec.
(AVI)

## Acknowledgments

We thank Ulrike Herian, Stephanie Kallis, Marie Bartenschlager and Micha Fauth for excellent technical assistance and Fredy Huschmand for IT assistance. We thank Dr. Thomas Pietsch-mann at TWINCORE—Centre for Experimental and Clinical Infection Research (Hannover, Germany) for providing HEK293T-miR122 cells. We would like to acknowledge the microscopy support from the Infectious Diseases Imaging Platform (IDIP), headed by Vibor Laketa, at the Center for Integrative Infectious Disease Research (CIID, Heidelberg, Germany) and the University of Heidelberg Electron Microscopy Core Facility (EMCF, Heidelberg, Germany) headed by Dr. Stefan Hillmer. We thank Dr. Barbara Mueller, Djordje Salai and Thorsten Mueller (CIID, Heidelberg, Germany) for kindly providing the CLIPf, CLIP$^{ATTO590}$, and mScarlet constructs, respectively. Plasmids pTRE3G-NlucP and pCMV-Sport6-CD63-

pHluorin were kind gifts from Masaharu Somiya (Addgene plasmid # 162595) and D.M. Pegtel (Addgene plasmid # 130901), respectively. We are grateful to all members of the Molecular Virology unit for continuous stimulating discussions.

## Author Contributions

**Conceptualization:** Minh-Tu Pham, Ji-Young Lee, Ralf Bartenschlager.

**Data curation:** Minh-Tu Pham.

**Formal analysis:** Minh-Tu Pham, Christian Ritter, Roman Thielemann, Janis Meyer.

**Funding acquisition:** Karl Rohr, Ralf Bartenschlager.

**Investigation:** Minh-Tu Pham, Ji-Young Lee, Uta Haselmann.

**Methodology:** Minh-Tu Pham, Ji-Young Lee, Christian Ritter, Roman Thielemann, Charlotta Funaya, Vibor Laketa, Karl Rohr.

**Project administration:** Ralf Bartenschlager.

**Resources:** Charlotta Funaya, Vibor Laketa, Karl Rohr, Ralf Bartenschlager.

**Software:** Christian Ritter, Roman Thielemann, Charlotta Funaya, Vibor Laketa, Karl Rohr, Ralf Bartenschlager.

**Supervision:** Ralf Bartenschlager.

**Validation:** Minh-Tu Pham.

**Visualization:** Minh-Tu Pham, Christian Ritter, Roman Thielemann.

**Writing – original draft:** Minh-Tu Pham, Ralf Bartenschlager.

**Writing – review & editing:** Minh-Tu Pham, Ji-Young Lee, Christian Ritter, Roman Thielemann, Janis Meyer, Uta Haselmann, Charlotta Funaya, Vibor Laketa, Karl Rohr, Ralf Bartenschlager.

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
