## [Decision Letter · Decision Letter 0]

8 Feb 2023

Dear Prof. Bartenschlager,

Thank you very much for submitting your manuscript "Endosomal egress and intercellular transmission of hepatic ApoE-containing lipoproteins and its exploitation by the hepatitis C virus" for consideration at PLOS Pathogens. As with all papers reviewed by the journal, your manuscript was reviewed by members of the editorial board and by several independent reviewers. In light of the reviews (below this email), we would like to invite the resubmission of a significantly-revised version that takes into account the reviewers' comments.

The reviewers in general appreciated the technical soundness of the investigation, but raised important questions about additional controls, and requested clarifications regarding the reproducibility and interpretation of several key experiments. In particular, please provide additional confirmations requested by reviewers #1 and #2 that overexpressed tagged ApoE faithfully recapitulates the behavior of the endogenous protein. Please also address the concerns of reviewer #2 that double ApoE-CD63 positive structures in the recipient cells may form due to independent secretion and endocytosis events and thus may not represent the postulated co-secretion and co-transmission phenomenon. 

We cannot make any decision about publication until we have seen the revised manuscript and your response to the reviewers' comments. Your revised manuscript is also likely to be sent to reviewers for further evaluation.

Sincerely,

George A. Belov, PhD

Academic Editor

PLOS Pathogens

Alexander Gorbalenya

Section Editor

PLOS Pathogens

Kasturi Haldar

Editor-in-Chief

PLOS Pathogens

orcid.org/0000-0001-5065-158X

Michael Malim

Editor-in-Chief

PLOS Pathogens

orcid.org/0000-0002-7699-2064

Reviewer's Responses to Questions

**Part I - Summary**

Reviewer #1: In this paper, the authors analyzed the egress of apolipoprotein E (ApoE) and its potential exploitation by hepatitis C virus (HCV). ApoE is synthesized at the endoplasmic reticulum and transported to the Golgi where it associates with lipoproteins. The authors focused their work on the transport of ApoE-lipoproteins between the Golgi and the plasma membrane. For this, they developed a fully functional fluorescently tagged ApoE which allows to follow the trafficking and egress of ApoE in hepatocytic cells. They showed that ApoE-lipoproteins accumulate in CD63 positive endosomes. Their data suggest co-secretion and cell-to-cell co-transmission of ApoE and endosome-derived extracellular vesicles. They also showed intracellular enrichment of HCV NS5A in ApoE-positive structures. They also observed the presence of NS5A- and ApoE-containing intraluminal vesicles in endosomes. Finally, their data suggest co-secretion and co-transmission of ApoE-positive lipoproteins with endosome-derived extracellular vesicles containing HCV NS5A and viral RNA. Globally, this work is technically sound. However, it remains rather descriptive. In particular, the localization of proteins in late endosomes does not necessarily mean that they are transported from the Golgi to this compartment. Indeed, large amounts of material ends up in late endosomes after being taken up from the extracellular medium or by autophagy. It is therefore needed to exclude these hypotheses to confirm the conclusions.

Reviewer #2: In this manuscript, Pham et al. aim at characterizing the intracellular trafficking of ApoE and understanding how ApoE egress is hijacked by HCV. To achieve these goals, the authors designed a fully functional FP-tagged ApoE and employed an integrative imaging approach to monitor the endosomal post-Golgi trafficking egress and transmission route of hepatic ApoE-LPs in uninfected cells and in the context of HCV infection.

The manuscript contributes to understanding the role of ApoE in cell biology and in viral infection. Moreover, the newly designed fusion protein represents an improved tool for tracking ApoE and monitoring its trafficking in normal and disease states. The novelty of the work is the discovery that ApoE-LPs co-egress with CD63-positive extracellular vesicles (EVs) via late endosomal trafficking and are transmitted from cell to cell. Moreover, HCV NS5A binding to ApoE is required for the release of these ApoE-associated CD63-positive EVs containing viral RNA that are transmitted between cells. Adequate controls were used and the conclusions are supported by the data.

Nevertheless, a few experiments/analyses are proposed to strengthen the conclusions, and major editing is required to improve the clarity.

Reviewer #3: The authors have developed a new tool to study ApoE secretion based on a fusion protein of ApoE with mTurquoise2. This tool allowed them to follow ApoE secretion in a hepatic cell line and to suggest that ApoE-containing lipoproteins (ApoE-LPs) follow the secretion pathway of CD63-positive late endosomes. Moreover, the authors showed that NS5A, a non-structural protein of hepatitis C virus (HCV), is enriched in ApoE-LP and follows the same late endosomal pathway for egress, and that ApoE-NS5A interaction is required for efficient release of extracellular vesicles containing the viral RNA genome. The authors have used several advanced microscopy technics such as live imaging, super-resolution microscopy, correlative light and electron microscopy (CLEM) as well as single molecule fluorescent in situ hybridization (smFISH).

The experiments are in general well conducted, but some of them lack repetitions or controls.

**Part II – Major Issues: Key Experiments Required for Acceptance**

Reviewer #1: 1. In Huh-7 cells, apoE-mT2 is observed as dots by immunofluorescence, which are localized in the secretory (PDI, GM130) and endocytic (CD63, rab7) pathways. However, it is not clearly demonstrated where the endosomal apoE-mT2 originates from. Is it transported from the Golgi to endosomes, or is it secreted in the medium and then re-internalized by endocytosis to reach endosomes?

2. What is the intracellular localization of apoE-mT2 in HeLa and 293T cells? Secretory or endocytotic pathways or both, like in Huh-7 cells? Is it secreted associated to lipoproteins in these non-hepatic cell lines, which do not express apoB?

3. It would be interesting to analyze the intracellular localization of apoE-mT2 in cells which do not internalize lipoproteins by endocytosis.

4. Fig 3, the argument of co-secretion and co-transmission is rather weak. Both structures could very well be secreted independently and associate in the extracellular medium or in the endocytic pathway of recipient cells. The authors should quantify ApoE+CD63- spots and apoE-CD63+ spots in recipient cells in addition to double positive spots. In addition, it would be interesting to study the effect of exosome secretion inhibitors. Would they inhibit ApoE and CD63 release to the same extent?

5. Fig 6, the NS5A-ApoE spots appear associated to double membrane vesicles in EM. Could these vesicles be replication complexes addressed to endosomes by autophagy? Would NS5A be directed to endosomes in autophagy-deficient cells? What about NS5A expressed alone? Does it co-localize with ApoE in CD63-positive endosomes?

Reviewer #2: Major comments:

1. Figure 1. a. There is no definition of naïve control – is this mock transfection?

b. In 1C and some of the other figures (e.g. 2A) the y-axis title says "normalized fluorescence intensity", however, it seems that raw values are plotted. If indeed normalized, what is it normalized to?

c. Figure 1D – How do the authors explain the finding that mT2 ApoE rescued the various phenotypes more than WT ApoE? The expression levels of the two proteins should be shown.

2. Figure 2. a. Figure 2D: An experiment showing the microtubule-dependent trafficking of Apo-CD63 endosomes would be an interesting addition to the paper.

b. Figure 2E (upper panel) depicts a single particle. What is the abundance of the ApoECD63pHluorin+ve particles?

3. Figure 3. a. Figure 3B. The morphology of the gold-ApoE vesicles is disrupted in contrast to the CD63+ vesicles. Can the authors speculate/elaborate on what might lead to the distortion of the vesicles during gold-ApoE vs. gold-CD63 labeling? b. On a related note, only large vesicles are positive for CD63, as shown in the bottom panel. What distinguishes the larger vesicles stained positive for CD63 from the small vesicles (except size)? It would be important to determine the fraction of CD63-positive EVs in secreted ApoE.

c. Figure 3E. How were the numbers of foci quantified? Using what software? Further details should be provided.

4. Figure 4. Line 341-342: “In agreement with a previous report (60), we observed time-dependent secretion of NS5ANluc into the cell culture supernatant (Fig 4F).” There is no Fig 4F in the manuscript.

5. Figure 7. a. Figure 7B. The Western blot of the same samples with anti-NS5A antibodies is missing and so is the immunoblot on whole cell lysate.

b. A clearer explanation regarding the TEM findings shown in Figure 7C will help the readers.

c. Line 370-372: “Remarkably, we could detect distinct foci of HCV RNA in single recipient cells, around 13% of them being ApoE-CD63 double-positive (example image in Fig S5C, area 2; quantification in Fig 7F).” Is 13% biologically significant? What is the basal level? More details should be provided about this quantification.

6. The conclusion that CD63/endosomal pathway mediate ApoE egress would be strengthened by showing that perturbation of this factor/pathway impacts ApoE egress (either in naïve of HCV-infected cells).

Reviewer #3: Major points:

1) The assumption that over-expressed tagged proteins are fully functional is not always well shown:

a) In Fig 1B, authors should show the fractionation of a supernatant from parental Lunet cells to appreciate densities of the different over-expressed ApoE forms with regards to endogenous ApoE.

b) In Fig S4C, authors should add ApoEmT2 to control that secretion levels between ApoESNAPf and ApoEmT2 are identical because the presented data only show that ApoESNAPf is secreted.

c) In Fig S4D, authors should use a standard replication assay to show that NS5ACLIPf is functional for HCV genome replication.

2) Fig 7 and its related Fig S5 suffer from lack of repetition and controls:

a) Fig 7A: Since there are 8 dots, I assume that there are 4 replicates in each of the two experiments, which seems to indicate a high variability. Please, indicate which dots come from which experiment since otherwise, the representation of the replicate is useless and might induce misunderstanding for the readers. I would also strongly suggest increasing the number of experiments to have at least 3 independent repeats.

Also indicate which set of data was used for the statistical analysis? i.e., on internal replicates or on the mean of the 2 experiments?

What is the level of total secreted HCV RNA in these cells? The authors should add the values of non-immuno-captured HCV RNA which represents the input fraction.

b) Fig 7B: Same as in Fig 7A, what are the values of the input fraction for NS5A (upper graph) or ApoE (lower WB)?

c) The authors should study the co-secretion of ApoE and HCV EVs from the same material. Indeed, the secretion level of HCV EVs containing HCV RNA and/or NS5A might not be the same in subgenomic HCV replicon cells (i.e., cells that stably replicate the HCV genome (Fig 7A)) than in HCV RNA transfected cells (Fig 7B).

d) Fig 7D: I have the same concerns regarding the representation of the replicates in this panel as in Fig 7A (see above comments 2a).

e) Fig S5B: Please show at least one entire cell with smFISH staining, not only a zoomed area.

f) Fig S5C: Please show entire cells. There is a strong red signal in the center of recipient cells: does it correspond to the background announced in line 363? If yes, how can the author discriminate background from real positive RNA signal? In line with this, HCV RNA dots in recipient cells appear to be much smaller than those in donor cells? How can the authors explain this as each single dot should represent one single RNA molecule?

3) In the Materials &Methods, it is stated that cells were treated with proteinase K after permeabilization, thus, how can the authors detect mCherry, mTurquoise2 and YFP signals?

Finally, how to be sure that the ApoE-HCV RNA-CD63 dots are inside the cell and not apposed on the plasma membrane of the recipient cells? The authors may try to block cell entry using ApoE and/or CD63 antibody or block the exosomal secretion of donor cells with the GW4869 inhibitor and include the results in Fig 7F.

**Part III – Minor Issues: Editorial and Data Presentation Modifications**

Reviewer #1: No minor point

Reviewer #2: Minor comments:

1. Materials and Methods are missing descriptions of the methods used for EV preparation and quantification of HCV core protein.

2. Figure 5B (and legend): consider using consistent representation of CLIP-ATTO590 and SNAP-SIR647 in the figure and text. (For ex: In line 590, superscript in all other places).

3. Figure 4: Use Either h or h.p.e. in C and E for consistency.

4. Line 208: "exclusively fluoresces" instead of "is exclusively excited"

5. Line 262: “spinning disc” should be spinning-disk confocal microscopy

6. Line 272: 10 sec/frame – clarify if this means each frame was taken with a 10 s exposure or whether 1 frame was taken every 10 s, in which case 1 frame/10 s may be more appropriate.

7. Line 601: Which Alexa fluorophore?

8. STED should be spelled out.

9. Consider rephrasing the following sentences for improved clarity:

Lines 66-69: These alterations…apolipoprotein E (ApoE).

Lines 148-150: Of note, ApoE puncta…live-cell imaging (Fig 1C).

Lines 331-333: In the first set of experiments...ApoE-specific pull-down.

Lines 403-405: For the poliovirus…apply to NS5A (73-75).

10. Figure 3D: The upper image missing a scale bar.

Reviewer #3: Minor points:

1) Fig 1A-B: It is not clear whether ApoE is over-expressed in naïve cells or not. Please clarify this point in the text and in figures.

When wt ApoE is over-expressed in naïve cells, there is a discrepancy in ApoE levels for naïve supernatants between Fig 1A and Fig 1B (the input level of ApoE in supernatant of naïve cells appears to be equal to ApoEmT2 supernatant level in Fig 1B, while there is a clear difference in Fig 1A). Which blot is representative of a standard experiment?

The authors should show the parental Lunet and Lunet-ApoE KD cells to appreciate the KD efficiency of ApoE in Fig 1A.

2) In the legend of Fig S1C-D: The authors wrote in line 1310 “Data are means from a representative experiment (n=3)”. Do they mean that the presented data are the means of internal replicates? Please clarify this point.

3) Fig 4C: The authors mentioned in lines 270-271 that NS5AmCherry-ApoEmT2 foci abundance increased “much higher as compared to NS5AmCherry-E2eYFP positive foci”, albeit they do not show quantification of the latter. Please show the quantification of NS5A-E2 positive foci. The interpretation of these quantifications should be more cautious since the HCVtcp system do not allow a huge assembly level of infectious virions, as suggested by the authors in lines 267-268.

4) Line 342, Fig 4F does not exist and should be replaced by Fig S4F.

5) Line 371-372: Do the authors mean “HCV RNA dots ApoE-CD63 double positive” or “HCV RNA ApoE positive” since in Fig 7F only quantification of HCV RNA dot that are ApoE positive is shown?

PLOS authors have the option to publish the peer review history of their article (what does this mean?). If published, this will include your full peer review and any attached files.

Reviewer #1: No

Reviewer #2: **Yes: **Shirit Einav

Reviewer #3: No
---

## [Decision Letter · Decision Letter 1]

7 Jul 2023

Dear Prof. Bartenschlager,

We are pleased to inform you that your manuscript 'Endosomal egress and intercellular transmission of hepatic ApoE-containing lipoproteins and its exploitation by the hepatitis C virus' has been provisionally accepted for publication in PLOS Pathogens.

Best regards,

George A. Belov, PhD

Academic Editor

PLOS Pathogens

Alexander Gorbalenya

Section Editor

PLOS Pathogens

Kasturi Haldar

Editor-in-Chief

PLOS Pathogens

orcid.org/0000-0001-5065-158X

Michael Malim

Editor-in-Chief

PLOS Pathogens

orcid.org/0000-0002-7699-2064

Reviewer Comments (if any, and for reference):

Reviewer's Responses to Questions

**Part I - Summary**

Reviewer #1: The revised version of the manuscript is clearly improved and I am satisfied with the modifications.

Reviewer #2: My comments have been addressed. This revised version is significantly improved.

Reviewer #3: The authors addressed all concerns, and this reviewer has no further comments to add.

**Part II – Major Issues: Key Experiments Required for Acceptance**

Reviewer #1: NA

Reviewer #2: None

Reviewer #3: (No Response)

**Part III – Minor Issues: Editorial and Data Presentation Modifications**

Reviewer #1: NA

Reviewer #2: None

Reviewer #3: (No Response)

PLOS authors have the option to publish the peer review history of their article (what does this mean?). If published, this will include your full peer review and any attached files.

Reviewer #1: No

Reviewer #2: No

Reviewer #3: No

---

## [Editor Report · Acceptance letter]

22 Jul 2023

Dear Prof. Bartenschlager,

We are delighted to inform you that your manuscript, "Endosomal egress and intercellular transmission of hepatic ApoE-containing lipoproteins and its exploitation by the hepatitis C virus," has been formally accepted for publication in PLOS Pathogens.

Best regards,

Kasturi Haldar

Editor-in-Chief

PLOS Pathogens

orcid.org/0000-0001-5065-158X

Michael Malim

Editor-in-Chief

PLOS Pathogens

orcid.org/0000-0002-7699-2064